# Control of replication and gene expression by ADP-ribosylation of DNA in *Mycobacterium tuberculosis*

Rachel E Butler[1], Marion Schuller [ID][2], Ritu Jaiswal [ID][3], Jayanta Mukhopadhyay[3], Jim Barber[1], Suzie Hingley-Wilson[1], Emily Wasson [ID][1], Alexassander Couto Alves[4], Ivan Ahel [ID][2] & Graham R Stewart [ID][1]

## Abstract

*Mycobacterium tuberculosis* maintains long-term infections characterised by the need to regulate growth and adapt to contrasting in vivo environments. Here we show that *M. tuberculosis* complex bacteria utilise reversible ADP-ribosylation of single-stranded DNA as a mechanism to coordinate stationary phase growth with transcriptional adaptation. The DNA modification is controlled by DarT, an ADP-ribosyltransferase, which adds ADP-ribose to thymidine, and DarG, which enzymatically removes this base modification. Using *darG*-knockdown *M. bovis* BCG, we map the first DNA ADP-ribosylome from any organism. We show that inhibition of replication by DarT is reversible and accompanied by extensive ADP-ribosylation at the origin of replication (OriC). In addition, we observe ADP-ribosylation across the genome and demonstrate that ADP-ribose-thymidine alters the transcriptional activity of *M. tuberculosis* RNA polymerase. Furthermore, we demonstrate that during stationary phase, DarT-dependent ADP-ribosylation of *M. tuberculosis* DNA is required to optimally induce expression of the Zur regulon, including the ESX-3 secretion system and multiple alternative ribosome proteins. Thus, ADP-ribosylation of DNA can provide a mechanistic link through every aspect of DNA biology from replication to transcription to translation.

**Keywords** ADP-ribosylation; PARP; DNA Modification; Transcription Regulation; ADPr-Seq
**Subject Categories** Chromatin, Transcription & Genomics; Microbiology, Virology & Host Pathogen Interaction; Post-translational Modifications & Proteolysis

## Introduction

*Mycobacterium tuberculosis* is the leading single cause of death by an infectious disease, killing 1.25 million people in 2023 (WHO, 2024). New antibiotics have recently been added to the multidrug panels that constitute the backbone of tuberculosis (TB) treatment.

Yet, *M. tuberculosis* still presents a frustrating recalcitrance to drug therapy, with treatment regimens typically lasting from 4 to 6 months even for fully drug-sensitive infections (WHO, 2022). Perhaps the biggest reason for this persistence in the face of efficacious drugs is the presence of subpopulations of mycobacteria which replicate slowly or not at all and are tolerant to drugs. In addition, a significant proportion of infected individuals carry a latent asymptomatic infection of putatively non-replicating mycobacteria, which can re-activate, perhaps years later, sustaining transmission. Indeed, the ability to tightly regulate replication and grow slowly may be an important trait for pathogenic bacteria in general (Leggett et al, 2017) and mycobacteria in particular.

ADP-ribosylation is a well-established posttranslational modification of proteins, but recently it became apparent that reversible ADP-ribosylation of nucleic acids is widespread (Groslambert et al, 2021; Suskiewicz et al, 2023). Previously, we defined the molecular mechanisms of an ADP-ribosyltransferase, DarT2 (hereon known as DarT), which catalyses DNA base modification by adding an ADP-ribose to the in-ring N3 of thymine in single-stranded DNA (ssDNA) (Schuller et al, 2021). In *M. tuberculosis*, DarT is co-expressed with the macrodomain-containing DarG protein, a DNA ADP-ribosylglycohydrolase, which regulates DarT activity by enzymatic removal of ADP-ribose from modified thymine in addition to direct antagonistic physical interaction with DarT (Deep et al, 2023; Jankevicius et al, 2016; Schuller et al, 2021).

DarT is present in many bacteria and, in some, an orthologous system, DarT1, is also present, catalysing ADP-ribosylation of guanosine bases in ssDNA (Cihlova et al, 2024; Schuller et al, 2023). In most bacteria, these systems appear to function in phage defence (LeRoux et al, 2022), however, the potential utility of reversible DNA modification raised the question of whether these enzyme systems may also provide mechanisms for regulation of DNA biology and cellular function. To answer this question, we chose to examine *M. tuberculosis* because DarT-DarG was ubiquitous in all members of the pathogenic *M. tuberculosis* Complex and was genomically situated in a transcriptional unit with the replicative helicase DnaB rather than in a phage defence island. We previously demonstrated that unregulated DarT activity was massively toxic to bacterial cells but the DarT-DarG module in *M. tuberculosis* regulated growth by ADP-ribosylation of the origin of chromosome replication (Schuller et al, 2021).

[1]Section of Bacteriology, School of Biosciences, University of Surrey, Guildford, Surrey GU2 7XH, UK. [2]Sir William Dunn School of Pathology, University of Oxford, Oxford OX1 3RE, UK. [3]Department of Chemical Science, Bose Institute, EN80 Sector V, Salt Lake, Kolkata, West Bengal 700091, India. [4]School of Human Development and Health, University of Southampton, Southampton, SO16 6YD, United Kingdom. ✉E-mail: ivan.ahel@path.ox.ac.uk; g.stewart@surrey.ac.uk

To investigate further the role of DarT-DarG in gene and chromosome regulation, here we examined the phenotypic effects of DarT-DarG activity on replication and investigated whether ADP-ribosylation of the mycobacterial genome provides a dynamically flexible switch. We utilised affinity-sequencing to identify ADP-ribosylation sites across the genome and explored how ADP-ribosylation of thymidine affects the initiation, regulation and progression of transcription, providing for the first time in any organism evidence that modification of a DNA base with ADP-ribose provides a mechanism to coordinate DNA replication with transcriptional adaptation.

# Results

## Regulation of mycobacterial growth by reversible ADP-ribosylation of DNA

Unregulated activity of the DarT ADP-ribosyltransferase is highly toxic to mycobacterial cells (Schuller et al, 2021; Zaveri et al, 2020), resulting in loss of viability equivalent to that achieved with some front-line antibiotics. Using an anhydrotetracycline (ATC)-inducible CRISPRi vector to knockdown DarG and deregulate DarT activity, we observed by fluorometry and live-cell microscopy that growth inhibition defined by plateau or reduction in viable colony counts resulted from inhibition of DNA synthesis (Figs. 1A and EV1A–C) and inhibition of cell division accompanied by elongation of cell length compared to control cells carrying the empty vector (Movies EV1 and EV2; Fig. EV2A–F). These elongated cells were demonstrated by scanning electron microscopy to have a nodular appearance (Fig. 1B) and were quantified by light microscopy of cells 7 days after DarG-knockdown to be approximately 3 times the length of control cells in which DarG-knockdown was not induced or which expressed non-specific sgRNA (Fig. 1B). Such filamentous cellular morphotypes are associated with perturbation of DNA replication or cell division in gene function screens (de Wet et al, 2020) and are naturally observed during infection of macrophages (Chauhan et al, 2006) or in sputum from TB patients (Garton et al, 2002; Ufimtseva et al, 2019; Vijay et al, 2017). To investigate if this growth inhibition by in cellulo ADP-ribosylation of DNA was reversible, we inhibited growth of *M. bovis* BCG for 7 days before washing away the ATC inducer and re-seeding fresh cultures with and without ATC to maintain DarG-inhibition or re-establish regulation of DarT activity. Re-establishing regulation of DarT allowed cells to resume growth whereas continued repression of DarG resulted in sustained growth inhibition (Fig. 1C). By live-cell microscopy, we observed that resumption of growth occurred when the arrested, elongated cells reinitiated septation although sometimes one daughter cell underwent necrosis (Figs. 1D and EV3A–I; Movie EV3). Combined with our previous demonstration of the in vitro addition and removal of ADP-ribose to thymine in ssDNA by DarT and DarG, respectively (Jankevicius et al, 2016; Schuller et al, 2021), this demonstrates that ADP-ribosylation of thymine can function as a reversible switch for growth regulation. This corroborates observations that the presence of DarT-DarG confers a slower growth rate on wild-type *M. tuberculosis* compared to an isogenic DarT-DarG-knockout mutant (Schuller et al, 2021).

## Genome-wide ADP-ribosylation of DNA by DarT

We have previously shown that DarT targets thymine bases in ssDNA for ADP-ribosylation in a sequence-specific manner

(Jankevicius et al, 2016; Schuller et al, 2021). DarT from *Thermus aquaticus* has an in vitro preference for the thymine in position 3 of the sequence TCTC; *Escherichia coli* and *M. tuberculosis* preferentially target TTT/TCT (Lawaree et al, 2020) and TTTW, respectively. DarT is able to target other sequences to a lesser extent but the absolute requirement appears to be a thymine at positions 1 and 3. There are approximately 11,700 occurrences of TTTW in the *M. tuberculosis/M. bovis* genome and greater than 200,000 TNT sequence motifs. To map the potential for DarT to ADP-ribosylate thymine throughout the genome, we developed a technique, ADPr-Seq, using an antibody reactive to ADP-ribosylated ssDNA (Fig. 2A) to pull-down modified DNA before identification of the enriched sequences by sequencing. ADP-ribosylation was induced by CRISPRi-knockdown of DarG in *M. bovis* BCG and preliminary experiments indicated that fragmentation by sonication (Fig. EV4A) or by various proprietary nucleases (Fig. EV4B) reduced sensitivity of the technique but fragmentation by restriction digest allowed pull-down of modified DNA (Figs. EV4C and EV5). We analysed the relative abundance of ADP-ribosylated DNA fragments using a negative binomial generalised linear regression of read counts to estimate the relative enrichment of ADP-ribosylated fragments in the induced DarG-knockdown mycobacteria versus uninduced mycobacteria.

We showed that 72 DNA fragments were significantly enriched in DarG-knockdown *M.bovis* BCG and alignment to the genome provided a gDNA ADP-ribosylome map indicating that this DNA modification occurred widely across the genome although with a pronounced concentration around the origin of replication (Fig. 2B; Dataset EV1). We have previously shown that ADP-ribosylation occurs at the AT-rich region of OriC where replication is initiated, and our findings here corroborate this and additionally show that ADP-ribose is found in a wider region of ~20 kb around OriC (Fig. 2B,C). This region is rich in genes encoding DNA replication and cell division proteins and we demonstrate ADP-ribosyl modifications in genes encoding DnaA, DnaN, RecF and DNA gyrase as well as the partition proteins ParA and B. Some of these genes are also important in the DNA damage and repair (DDR) response and notably many other DDR genes, located in the wider genome, incorporated DNA regions that were enriched with ADP-ribose modifications (Fig. 2D) including single-stranded binding protein (*ssb*), two DNA helicases (Lhr and Mb3226c), the DNA-damage-inducible *dnaQ-uvrC* paralogue Mb2214c, the *mreII* dsDNA break repair gene, and the Mfd transcription-coupled repair gene. We also observed significant enrichment of DNA regions within metabolic genes, cell wall synthesis, and metal transport and utilisation genes (Fig. 2D).

ADP-ribosylation of thymidine nucleosides by DarT is restricted by DNA target sequence and also the occurrence and accessibility of ssDNA. The OriC itself has multiple high-efficiency DarT target sequences in the AT-rich region which functions primarily as the initial site for ssDNA generation allowing loading of the DnaB helicase (*co-transcribed with darTdarG*) and the DNA polymerase which further unwind and replicate DNA. It is thus clear that this region has the biochemical and physiological characteristics that explain the high level of ADP-ribosyl modification observed. We postulate that ADP-ribosylation of thymines outside of the OriC region may be partly dependent on transcriptional activity to provide the ssDNA template for DarT.

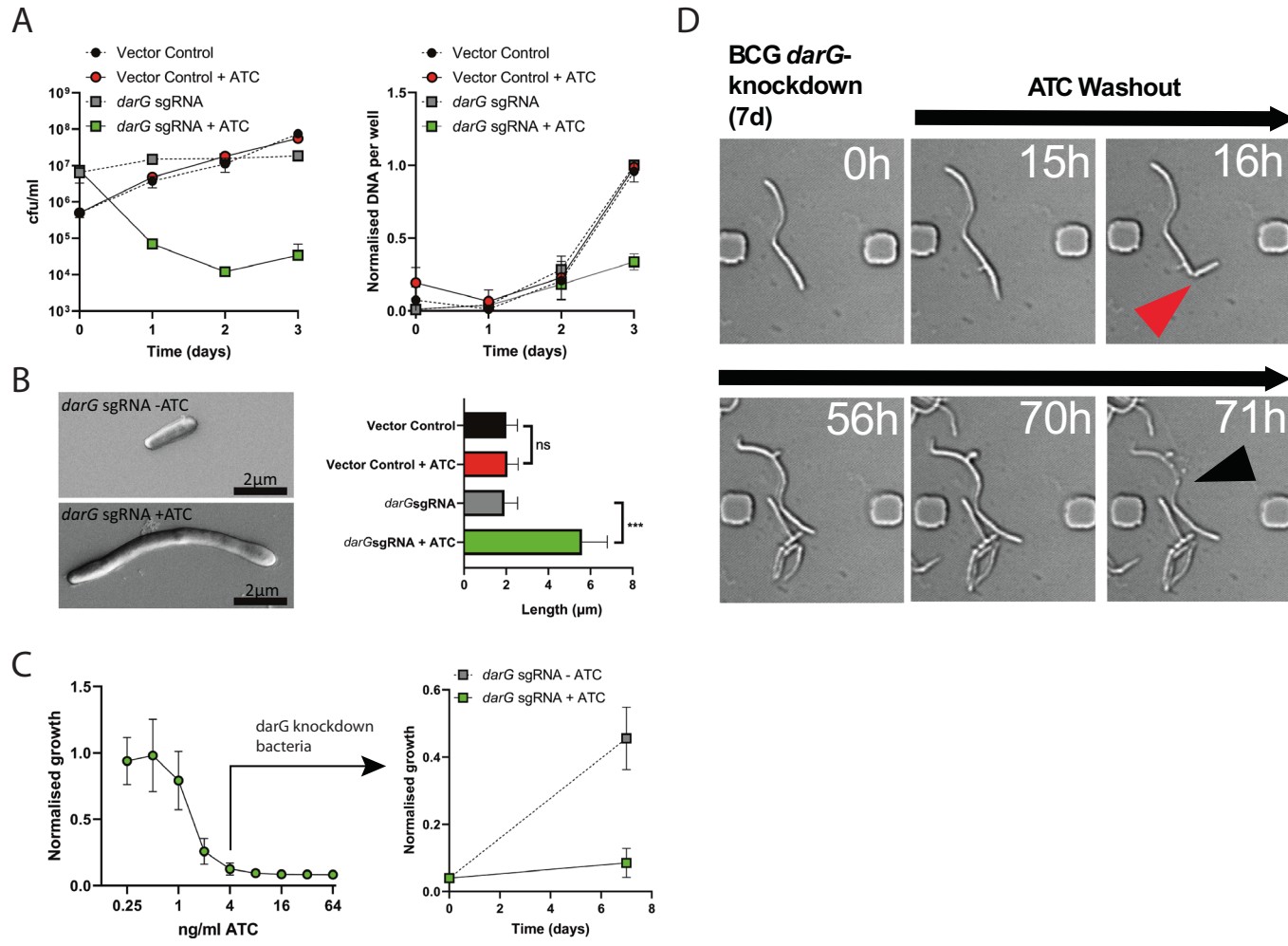

**Figure 1. Reversible control of mycobacterial replication by the DarT/DarG ADP-ribosylation system.**

(A) Left panel: Viability of *M. bovis* BCG is reduced when DarT activity is deregulated by CRISPRi knockdown of the "antitoxin" DarG ADP-ribosylglycohydrolase, induced by addition of anhydrotetracycline (ATC) at 200 ng/ml to *M. bovis* BCG *darG*-sgRNA. Right panel: DNA synthesis is inhibited by deregulation of DarT. Data are mean ± SD, *n* = 3 independent cultures. (B) Scanning electron micrograph showing filamentous appearance of BCG *darG*-knockdown mycobacteria with unregulated DarT ADP-ribosylation. Average cell length increased threefold in BCG *darG*-knockdown. Data are mean ± SD, *N* = 3 independent experiments, ~350 cell measurements per strain/treatment, ***P* = 0.0008 by one-way ANOVA with Sidak's multiple comparison test. (C) Level of growth inhibition is controllable by titration of ATC-induced knockdown of DarG and reversible by washout of ATC. Right panel shows *M. bovis* BCG *darG*-sgRNA inhibited for growth with 4 ng/ml ATC, washed and then restimulated with ATC or carrier alone. Data are mean ± SD, *n* = 3 independent cultures, and are representative of three independent experiments. (D) Captured images from video of a *darG*-sgRNA (growth inhibited) *M. bovis* BCG after washout of ATC showing resumption of cell division (red arrow), and eventual necrosis of one daughter cell (black arrow). The full video and selected stills are available in Movie EV3 and Fig. EV3. Source data are available online for this figure.

## ADP-ribosylation of DNA modulates transcription by the mycobacterial RNA polymerase

To explore the potential for DarT-dependent ADP-ribosylation of DNA to provide a regulatory mechanism for transcription, we investigated the in vitro capacity of DarT to modify DNA in the canonical ribosomal RNA *rrnAP3* promoter of *M. tuberculosis*. Using overlapping oligonucleotides representing the template and non-template strands of the *rrnAP3* promoter we showed that *M. tuberculosis* DarT added ADP-ribose to thymines in the promoter sequence with a preference for the thymine-rich region located on the non-template strand immediately proximal (5') to the −10 region (Fig. 3A). There was less efficient modification of the

template strand where the only possible TNT sequence was in the −10 region, and very low-level modification of the non-template strand at the −35 region (Fig. 3A). Thus ADP-ribosylation occurs in the promoter regions which come into direct contact with σA of the RNA polymerase holoenzyme.

To understand how ADP-ribosylation of thymines in the *rrnAP3* promoter affected transcription, we performed in vitro transcription (IVT) using the *M. tuberculosis* RNA polymerase holoenzyme and dsDNA with and without ADP-ribosylation of the non-template strand which includes the highly modified region adjacent to the −10. ADP-ribosylated ssDNA and annealed dsDNA IVT templates are shown in Fig. 3B. We also tested if ADP-ribosylation of the template strand affected transcription, this time

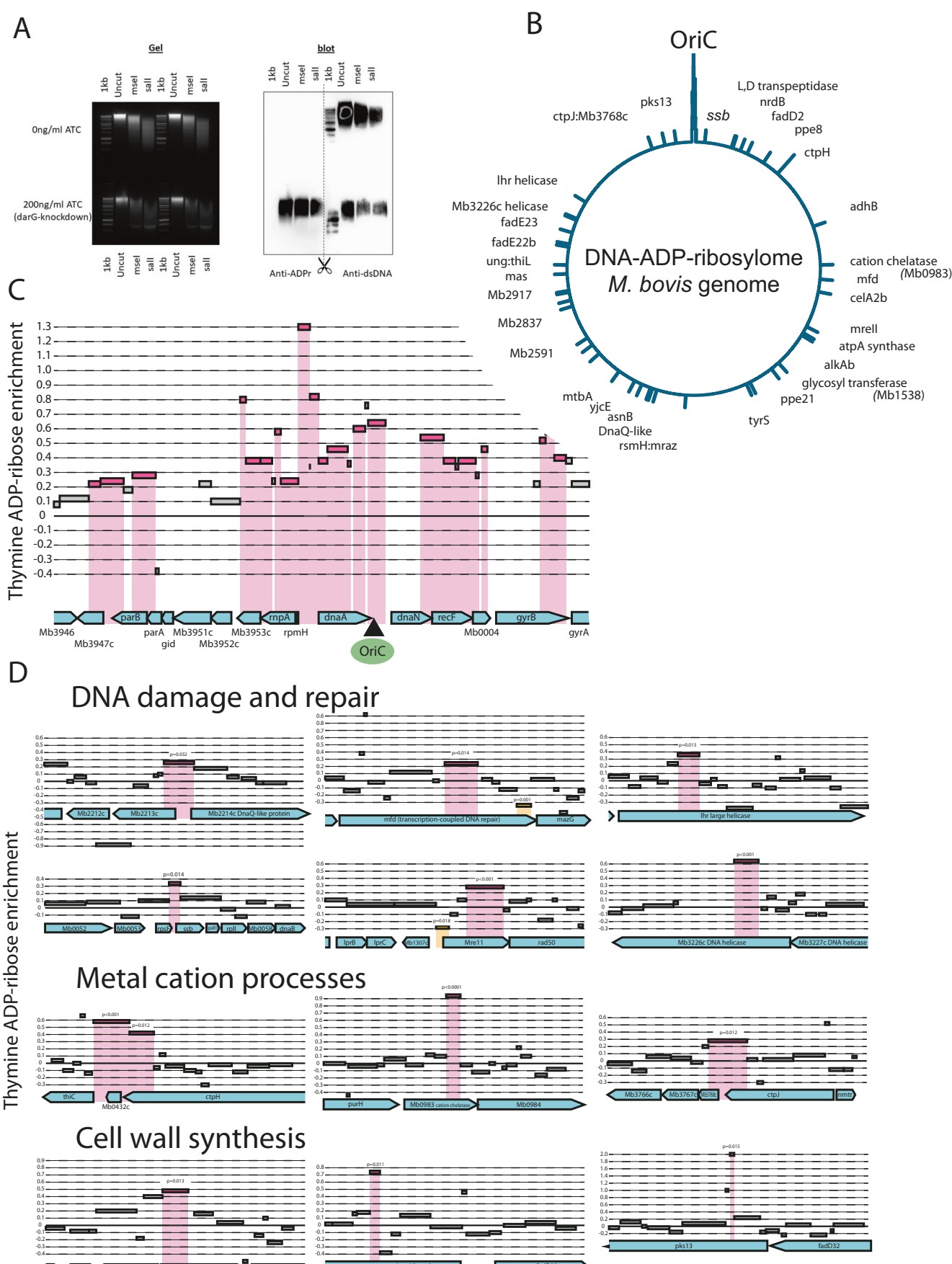

**Figure 2.  The DNA ADP-ribosylome of *M. bovis* BCG.**

(A) DarT-dependent ADP-ribosylation of DNA in *M. bovis* BCG *darG*-sgRNA was stimulated by 200 ng/ml ATC-induced knockdown of *darG*. DNA from ATC-induced and uninduced cells was Southern blotted and probed with antibodies against ADP-ribosylated DNA, indicating widespread DNA modification in DarG-knockdown cells. Anti-dsDNA blotting as control for DNA transfer. (B) ADPr-Seq of DNA from *darG*-knockdown *M. bovis* BCG. Alu1-digested DNA from ATC+ and ATC- *M. bovis* BCG *darG*-sgRNA was sequenced before and after immune-affinity capture of ADP-ribosylated DNA fragments. Significant enrichment of ADP-ribosylated DNA fragments was revealed by generalised linear regression of read counts to estimate the relative enrichment of modified fragments in the induced DarG-knockdown mycobacteria compared to control uninduced bacteria. The density of ADP-ribosylated DNA (FDR ≤0.05, n = 3)) is plotted across the *M. bovis* genome. (C) ADP-ribosylation of the chromosome origin of replication (OriC), including the AT-rich region and surrounding genes. Significantly enriched (ADP-ribosylated) ADPr fragments determined by negative binomial generalised linear regression of read counts to estimate the relative enrichment of ADPr fragments in the induced DarG-knockdown mycobacteria versus uninduced mycobacteria (FDR ≤0.05, n = 3) are shown in pink. (D) ADP-ribosylated DNA was distributed across the genome in genes associated with functional roles including DNA damage and repair, metal ion transport and biology and cell wall synthesis. Examples of ADP-ribose-enriched fragments are shown (FDR ≤0.05, n = 3). Source data are available online for this figure.

using an altered IVT substrate (*rrnAP3+*) with a single base change (g to t at +23) in the template strand which resulted in the incorporation of several preferred *M. tuberculosis* DarT modification sites in the transcribed region between +19 to +22. IVT of the non-ADP-ribosylated *rrnAP3/rrnAP3+* promoter produced a relatively high level of abortive transcripts but also some 47 nt run-off transcript. ADP-ribosylation of the non-template strand almost completely inhibited the synthesis of both abortive transcripts as well as run-off transcript (Figs. 3B and EV6). IVT of the *rrnAP3+* DNA carrying ADP-ribosylation of the template strand exhibited no inhibition of the synthesis of abortive transcripts, although there was some evidence for stalling of the elongation complex near the introduced ADP-ribosylation sites (Figs. 3B and EV6). The latter requires further investigation.

We further investigated the effect of ADP-ribosylation of the template strand of *rrnAP3* on the interaction with RNA polymerase using electromobility shift assay (EMSA). The result demonstrated that ADP-ribose modification did not inhibit the binding of the RNA polymerase holoenzyme to the promoter DNA (Fig. 3C). Whether the observed inhibition of RNA transcript production from templates with ADP-ribosylation of the non-template strand adjacent to the −10 and −35 is due to an inability to form the open complex or an inability of the RNA polymerase to escape from the ADP-ribose modified promoter needs further investigation because a heparin resistant shift is indicative but not proof of open complex formation.

## ADP-ribosylation of DNA regulates transcriptional adaptation to the stationary phase

With the knowledge that *darTdarG* is transcriptionally linked to the replicative helicase DnaB and is able to ADP-ribosylate (i) the OriC to control growth, and (ii) ssDNA in other areas of the genome, including actively transcribed genes, we hypothesised that the DarT-DarG module may provide a mechanism to coordinate gene expression with the replicative state of *M. tuberculosis*. To test our hypothesis, we compared genome-wide transcription levels between wild-type *M. tuberculosis* and a *darTdarG*-knockout strain, *M. tuberculosis* Δ*darTdarG*, during logarithmic and stationary growth phases. During logarithmic growth (6 day culture) no genes were expressed significantly differently between the strains but in stationary phase (20 days), 29 genes were significantly downregulated in the *darTdarG*-knockout strain compared to the wild-type and 4 genes were upregulated (Fig. 4A; Dataset EV2). The downregulated genes comprised 21 members of the *zinc uptake*

*regulator* (Zur) regulon (Maciag et al, 2007), including 5 alternative ribosomal proteins, 11 genes encoding the ESX-3 Type VII secretion and the zinc chaperone YciC. The Zur regulon of *M. tuberculosis* is induced in zinc-limiting conditions (Maciag et al, 2007) as well as in stationary phase (Voskuil et al, 2004), sputum (Lai et al, 2021) and in antibiotic persisters (Keren et al, 2011). Several genes not known to be Zur-regulated were also downregulated in the Δ*darTdarG* mutant compared to the wild-type alongside non-coding RNAs, notably MTS2823 which interacts with RNA polymerase enzyme and is upregulated in the stationary phase.

Zur acts as a transcriptional repressor binding to DNA in promoter regions with a conserved AT-rich palindromic target sequence in *M. tuberculosis* (Maciag et al, 2007). We observed that all *M. tuberculosis* Zur-binding sites contain a preferred DarT ADP-ribosylation motif on one or both DNA strands. We showed that *M. tuberculosis* DarT can transfer ADP-ribose to these Zur-binding sites (Fig. 4B) and then we determined by EMSA that binding of the Zur repressor to either of the Zur-binding sites located in the divergent promoter between *rpmB2* and Rv2059 is inhibited by ADP-ribose modification of either DNA strand (Fig. 4C). This was consistent with a scenario where ADP-ribosylation of Zur-binding sites during stationary phase inhibits binding by the Zur protein and consequently derepresses transcription. An alternative explanation was that the observed differences in Zur-regulon expression were due to differences in replication between the wild-type and DarT-DarG mutant strains without direct involvement of ADP-ribosylation at the differentially transcribed DNA. To control for this, we examined gene transcription in log-phase bacteria treated with the zinc chelator TPEN at levels which limited zinc but did not inhibit the growth of the mycobacteria. We observed that the *M. tuberculosis* Δ*darTdarG* was impaired in its ability to induce Zur-regulated genes although at levels significantly lower than observed in the stationary phase (Fig. 4D; Dataset EV2). Taken together, these observations are consistent with a model where DarT-dependent ADP-ribosylation of Zur-binding sites during stationary phase growth affords regulation of the Zur regulon by inhibiting binding of Zur, allowing derepressed transcription from the downstream promoter.

## Discussion

The ability to grow slowly and to carefully coordinate replication in synchrony with a host may be an important feature of many

A

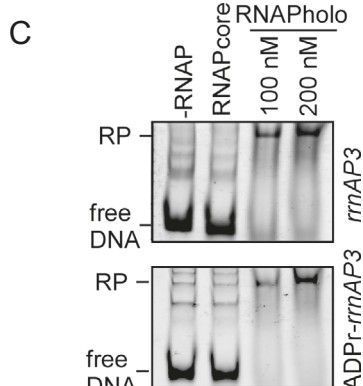

B

C

**Figure 3. The modulatory effects of ADP-ribosylation of ssDNA on RNA polymerase and transcription.**

(A) DarT from *M. tuberculosis* ADP-ribosylates the *rrnAP3* promoter in vitro. Overlapping oligonucleotides covering the promoter region on both template (T) and non-template (NT) strands were reacted with *M. tuberculosis* DarT and β-NAD$^+$, and assessed by PAGE to reveal ADP-ribose modifications. The strict TNT sequence requirement for DarT allowed inference of ADP-ribosylated thymidines (indicated by *) with preference for the non-template strand adjacent in the −10 region. (B) Transcriptional activity of RNA polymerase (RNAP) holoenzyme is inhibited by ADP-ribosylation of DNA. The NT strand of the *rrnAP3* promoter region (−59 to +47) was ADP-ribosylated with DarT enzyme. The T strand of the *rrnAP3+* promoter (carrying an g to t base change at +23) was ADP-ribosylated. ADP-ribose modified NT strand *rrnAP3* was annealed with the unmodified T strand to make *rrnAP3/ADPr* template DNA for in vitro transcription (IVT). ADP-ribose modified T strand of *rrnAP3+* was annealed with the unmodified NT strand to make *rrnAP3 + /ADPr* template DNA for IVT. IVT was performed with the *M. tuberculosis* RNAP holoenzyme on *rrnAP3/ADPr* template and *rrnAP3 + /ADPr* template alongside templates carrying no ADP-ribose modifications. This showed that ADP-ribosylation of the NT strand of *rrnAP3* inhibited the formation of both abortive transcripts as well as run-off transcripts, whereas ADP-ribosylation on the template strand of *rrnAP3+* did not inhibit transcription although there was some indication for formation of stalled elongation complex near the introduced ADP-ribosylation site. This IVT assay is shown alongside a replicate experiment in Fig. EV6. (C) EMSA showing RNAP holoenzyme binding to the *rrnAP3* promoter irrespective of ADP-ribose modification. Source data are available online for this figure.

obligate bacterial pathogens (Leggett et al, 2017). This may be particularly true for an intracellular pathogen like *M. tuberculosis*, whose life-strategy relies on chronic or dormant infection and balanced manipulation of the host immune response to drive transmission without stimulating its own immune destruction (Stewart et al, 2003). During the evolution of the *M. tuberculosis* complex, the bacterium acquired approximately 90 so-called toxin-antitoxin systems (Ramage et al, 2009; Sala et al, 2014). Why *M. tuberculosis* has retained so many of these systems has remained a matter of great conjecture but biochemical and physiological study of some suggests they provide mechanisms to regulate bacterial growth and adaptation in response to different environmental cues (Barth et al, 2021; Barth et al, 2019; Slayden et al, 2018). Previously, we characterised the molecular mechanism of one of these systems, DarT-DarG, for which DarT functions as an ADP-ribosyltransferase catalysing ADP-ribosylation of thymidine nucleosides in ssDNA, and DarG is a glycohydrolase that removes the DNA modification. We showed that deletion of this system conferred faster growth on the bacterium (Schuller et al, 2021) corroborating genome-wide fitness assays including one study utilising chemostat-controlled growth which demonstrated that without *darT*, *M. bovis* BCG was unable to grow at a slow replication rate (Beste et al, 2009). Here, we reveal the first DNA ADP-ribosylome from any organism; identifying the genomic sites of ADP-ribosylation in *M. bovis* BCG and highlighting pronounced ADP-ribose modifications at and around the OriC as the mechanistic basis for replication control.

In addition to the control of mycobacterial growth, we show that reversible ADP-ribosylation of DNA by DarT-DarG provides a way to link replication state of the bacterium to transcription, thereby coordinating gene expression with growth phase. Our biochemical data suggest two mechanisms in which gene regulation by ADP-ribosylated ssDNA may occur: (1) Inhibiting transcription initiation; (2) Inhibiting binding of transcription factors with chromosomal DNA (Fig. 5). In support of the latter mechanism, we show under normal physiological conditions, that DarT-catalysed ADP-ribose modification of thymine bases in the DNA binding sites for the Zur transcription repressor is necessary to maximally induce expression of the Zur regulon during stationary phase. Zur controls genes associated with metal homoeostasis including the ESX-3 Type-7 secretion system which is important for zinc and iron acquisition, and essential for pathogenesis (Maciag et al, 2007; Tufariello et al, 2016). In addition, Zur regulates the alternative ribosome proteins RpmB2 (L28), RpmG1 (L33), RpsN2 (S14) and RpsR2 (S18) which may provide ribosomal stability and antibiotic tolerance (Li et al, 2018), and/or the potential for an altered repertoire of translated proteins during stationary phase (Chen et al, 2020; Dow and Prisic, 2018; Prisic et al, 2015) or other growth states where alternative ribosome protein expression is induced including slow growth (Beste et al, 2007), sputum (Lai et al, 2021) and antibiotic persisters (Keren et al, 2011). Thus, ADP-ribosylation of DNA can provide a mechanistic link through every aspect of DNA biology from replication to transcription to translation.

The DarT-DarG ADP-ribosylation system occurs in all members of the *M. tuberculosis* Complex with the exception that it only occurs in some strains of *M. canettii* which represents the species most closely related to the common ancestor of the pathogenic tubercle bacilli. We speculate that acquisition of the DarT-DarG system and the added level of control that it provided to replication and gene regulation was critical in *M. tuberculosis* evolution to a pathogen with the ability to establish asymptomatic, latent infection characterised by a slow, carefully controlled growth state. The detailed mechanistic basis of how ADP-ribosylation of DNA is triggered in synchrony with replication remains to be fully elucidated, although the transcriptional linkage of DarT-DarG to the DnaB helicase probably forms an important aspect of the process. Understanding the mechanisms and pathogenic role of DarT-DarG may allow the development of new and urgently needed treatment strategies for tuberculosis. We furthermore predict that regulation by ADP-ribosylation of DNA occurs in other organisms and that the techniques and concepts described here will help reveal its wider role in biology.

# Methods

**Reagents and tools table**

| Reagent/resource | Reference or source | Identifier or catalogue number |
|---|---|---|
| **Experimental models** | | |
| *Mycobacterium bovis* BCG | Laboratory stocks | BCG/Pasteur 1173P2 |
| *Mycobacterium bovis* BCG pRH2502 (dCas9Spy) | Schuller et al, 2021 | |
| *Mycobacterium bovis* BCG pRH2502 pRH5982 (darG-sgRNA) | Schuller et al, 2021 | Table EV1 |

| Reagent/resource | Reference or source | Identifier or catalogue number |
|---|---|---|
| *Mycobacterium bovis* BCG PLJR965-DarG | This study | Table EV1 |
| *Mycobacterium bovis* BCG PLJR965-NT | This study | |
| *M. tuberculosis* GC1237 | Caminero JA et al (2001) https://doi.org/10.1164/ajrccm.164.7.2101031. PMID: 11673204. | |
| *M. tuberculosis* Δ*darTG* | Schuller et al, 2021 | |
| **Recombinant DNA** | | |
| 23AAPGMD_Zur_pET151/D-TOPO | This study | Invitrogen (Thermo Fisher Scientific) |
| PLJR965-DarG | This study | Rock et al, 2017 |
| PLJR965-NT | This study | Rock et al, 2017 |
| **Antibodies** | | |
| Poly/Mono-ADP-Ribose (E6F6A) | Cell Signaling Technology | #83732 |
| Goat anti-rabbit peroxidase | Merck | A6154 |
| Mouse autoanti-dsDNA | DHSB | AB_10805293 |
| Goat anti-mouse IgGFc peroxidase | Merck | A0168 |
| **Oligonucleotides and other sequence-based reagents** | | |
| Zur site oligos | This study | Table EV2 |
| rnaa promoter sequences | This study | Table EV2 |
| rnaa sites | This study | Table EV2 |
| Oligos for Zur EMSA | This study | Table EV2 |
| sgRNA oligos in pLJR965 | This study | Table EV2 |
| **Chemicals, enzymes, and other reagents** | | |
| CyQUANT™ Direct Cell Proliferation Assay | Thermo Fisher Scientific | C35011 |
| Middlebrook 7H11 Agar Base | Merck | M0428 |
| Middlebrook 7H9 Broth Base | Merck | M0178 |
| Thermo Scientific™ Remel™ OADC Enrichment | Thermo Fisher Scientific | R450605 |
| Middlebrook ADC Growth Supplement | Merck | M0553 |
| Anhydrotetracycline | Thermo Fisher Scientific | IBA 20401002 |
| Kanamycin sulphate from *Streptomyces kanamyceticus* | Merck | K1377 |
| Gibco™ Hygromycin | Thermo Fisher Scientific | Gibco™ 10687010 |
| Ampicillin sodium salt | Merck | A9518 |
| Kinyoun Carbol Fuchsin | Pro-Lab Diagnostics | PL.7021 |
| Pierce™ 16% Formaldehyde (w/v), Methanol-free | Thermo Fisher Scientific | 28908 |
| Lysing Matrix B | MP Biomedicals | SKU: 1169110-CF |

| Reagent/resource | Reference or source | Identifier or catalogue number |
|---|---|---|
| Phenol – chloroform – isoamyl alcohol mixture | Merck | 77617 |
| QuantiFluor® dsDNA System | Promega | E2671 |
| Alu1 | New England Biolabs | R0137 |
| Mse1 | New England Biolabs | R0525 |
| Sal1 | New England Biolabs | R0138 |
| Zeta-Probe® Membrane | Bio-Rad | #1620158 |
| Clarity Western ECL Substrate | Bio-Rad | #1705061 |
| SuperSignal™ West Pico PLUS Chemiluminescent Substrate | Thermo Fisher Scientific | 34577 |
| RNase, DNase-free | Roche | 11119915001 |
| Invitrogen™ SYBR™ Safe DNA Gel Stain | Thermo Fisher Scientific | S33102 |
| Genomic DNA Clean & Concentrator-10 | Zymo Research | D4010 |
| Eppendorf™ DNA LoBind™ Tubes | Thermo Fisher Scientific | Eppendorf™ 0030108051 |
| Dynabeads™ Protein A for Immunoprecipitation | Thermo Fisher Scientific | 10001D |
| Proteinase K, Molecular Biology Grade | New England Biolabs | P8107 |
| ChIP DNA Clean & Concentrator | Zymo Research | D5205 |
| RNAlater® | Merck | R0901 |
| TRI Reagent® | Merck | T9424 |
| RNA Clean & Concentrator-5 (DNase Included) | Zymo Research | R1013 |
| RNAse | QIAgen | |
| Agilent Technologies Agilent RNA 6000 Nano Kit | Agilent Technologies | 50671511 |
| TPEN, N,N,N′,N′-tetrakis-(2-Pyridylmethyl)ethylenediamine | Merck | 616394 |
| BL21(DE3)pLysS Singles™ Competent Cells | Merck | F- ompT hsdSB(rB-mB-) gal dcm (DE3) pLysS (CamR) Cat # 70236 |
| IPTG | Thermo Fisher Scientific | 34060 |
| B-PER™ Complete Bacterial Protein Extraction Reagent | Thermo Fisher Scientific | 89822 |
| Pierce™ Protease Inhibitor Tablets, EDTA-free | Thermo Fisher Scientific | A32965 |
| His GraviTrap™ TALON columns | Cytiva | 29000594 |
| TEV Protease | Merck | T4455 |
| PD MidiTrap G-25 | Cytiva | 28918008 |

| Reagent/resource | Reference or source | Identifier or catalogue number |
|---|---|---|
| Thermo Scientific™ HisPur™ Ni-NTA Resin | Thermo Fisher Scientific | 88221 |
| BugBuster® Protein Extraction Reagent | Merck | 70584-M |
| cOmplete™, EDTA-free Protease Inhibitor Cocktail | Roche | COEDTAF-RO |
| Benzonase® Nuclease | Merck | E1014 |
| Ni-Sepharose High Performance histidine-tagged protein purification resin | Cytiva | 17526801 |
| 30% Acrylamide/Bis-acrylamide 37.5:1, Polyacrylamide solution | Merck | A3699 |
| Invitrogen™ SYBR™ Gold Nucleic Acid Gel Stain (10,000X Concentrate in DMSO) | Thermo Fisher Scientific | S11494 |
| **Software** | | |
| NIS-Elements Advanced Research | Nikon Instruments Inc | |
| FIJI (Image J) | https://imagej.net/software/fiji/ | |
| Cytocam | Micropix | |
| **Other** | | |
| CellASIC ONIX plate for bacteria cells (4 chamber) | Merck | B04A-03-5PK |
| Illumina TruSeq Pair-Ended Sequencing | Azenta Genewiz | |
| Illumina HiSeq | Azenta Genewiz | |
| Whole Genome Sequencing (Short 2 ×250 bp Illumina Reads) | Microbes NG | |

## Bacterial strains and culture methods

*Mycobacterium bovis* BCG and *Mycobacterium tuberculosis* strain GC1237 (Caminero et al, 2001) were maintained on Middlebrook 7H11 solid medium (Merck) supplemented with 0.5% glycerol and 5% oleic acid-albumin-dextrose-catalase (Remel™, Thermo Scientific). Broth cultures were maintained in Middlebrook 7H9 (Merck) supplemented with 0.05% Tween-80, 0.2% glycerol, and 10% OADC (BCG) or 10% ADC (GC1237, BCG). Construction of *M. tuberculosis* GC1237 Δ*darTdarG* was previously described (Schuller et al, 2021). The wild-type *M. tuberculosis* GC1237 was derived from a single colony and passaged only twice in broth before transformation with the knockout construct. Wild-type and mutant strains were sequenced to verify the absence of secondary mutations. All strains and plasmids used in this study are shown in Table EV1.

## Knockdown of *darG* expression in *M. bovis* BCG

For ADPR-Seq experiments to identify ADP-ribosylation sites, and RNA-Seq experiments, *darG* was knocked down using the two-

plasmid system of pRH2502 (dCas9Spy) and *pRH2521-darG-sgRNA* (pRH5982) (*darG*-sgRNA) (as described (Schuller et al, 2021)). For experiments examining the reversibility of ADP-ribosylation in *M. bovis* BCG, the single vector CRISPRi plasmid pLJR965 (Rock et al, 2017) was used to inducibly express dCas9 from *Streptococcus thermophilus* and sgRNA targeting the non-template strand at the 5' end of *darG*. Construction involved annealing the oligonucleotides *DarG1-upper* and *DarG1-lower* (Table EV2) and ligation into BsmB1 cut PLJR965 to make PLJR965-DarG. A non-targeting control plasmid was generated by annealing *Non-target-upper* and *Non-target-lower* oligonucleotides (Table EV2) and cloning into PLJR965. Plasmids were maintained with kanamycin at 50 µg/ml in *E. coli* and 25 µg/ml in *M. bovis* BCG and induced with anhydrotetracycline (ATC) at various concentrations.

## Cell length measurements and morphology

*M. bovis* BCG cultures carrying the pRH2502 (dCas9Spy) and pRH5982 (*darG*-sgRNA) or an empty vector control were grown to log phase, adjusted to OD600 nm of 0.05, and induced with/without 200 ng/ml ATC for 7 days. Bacilli were heat fixed onto microscope slides and stained with Kinyoun Carbol Fuchsin stain (Prolab) according to the manufacturer's instructions, before mounting with a coverslip and 25% glycerol in water as mounting medium. A minimum of 39–60 bacilli were counted for each condition in $n = 2$–3 replicate cultures. The average (Mean) of three independent experiments is presented, representing 358, 352, 362, and 348 cell measurements respectively for the vector control, vector control +ATC, *darG-sgRNA* and *darG-sgRNA* + ATC strains/treatments. For SEM analysis of morphology, cultures were washed with PBS and fixed for 16 h with 4% paraformaldehyde in PBS at 4 °C. Samples were washed twice in distilled water, smeared onto Lifter Slips (Erie Scientific) and air-dried in a laminar flow hood. Samples were coated with 3 nm gold particles using a Quorum Q150 EV Plus sputter coater, and imaged using a Thermo Fisher Apreo 2 High Vacuum SEM and Pathfinder microanalysis software.

## DNA content measurement

*M. bovis* BCG cultures carrying the pRH2502 (dCas9Spy) and pRH5982 (*darG*-sgRNA) or an empty vector control were grown to log phase, adjusted to OD600 nm of 0.05, and induced with/without ATC for 3 days (200 ng/ml). DNA content was measured using the CyQuant Cell Proliferation Assay (Thermo Fisher Scientific) according to the manufacturer's instructions. Data were normalised using the formula $(x\text{-min}(x))/(\max(x)\text{-min}(x)) = $ normalised value, for each control vs ATC-treated set. In a separate series of experiments, DNA was purified from ATC-treated/untreated *darG*-sgRNA mycobacteria by bead beating and phenol:chloroform extraction. Total DNA content of the cultures was measured using the dsDNA Quantifluor system (Promega) and compared to OD600 nm and CFU/ml measurements of the cultures.

## ADPR-seq

*M. bovis* BCG bacilli carrying the plasmids pRH2502 (dCas9Spy) and pRH5982 (*darG*-sgRNA) were grown to mid log phase in 7H9

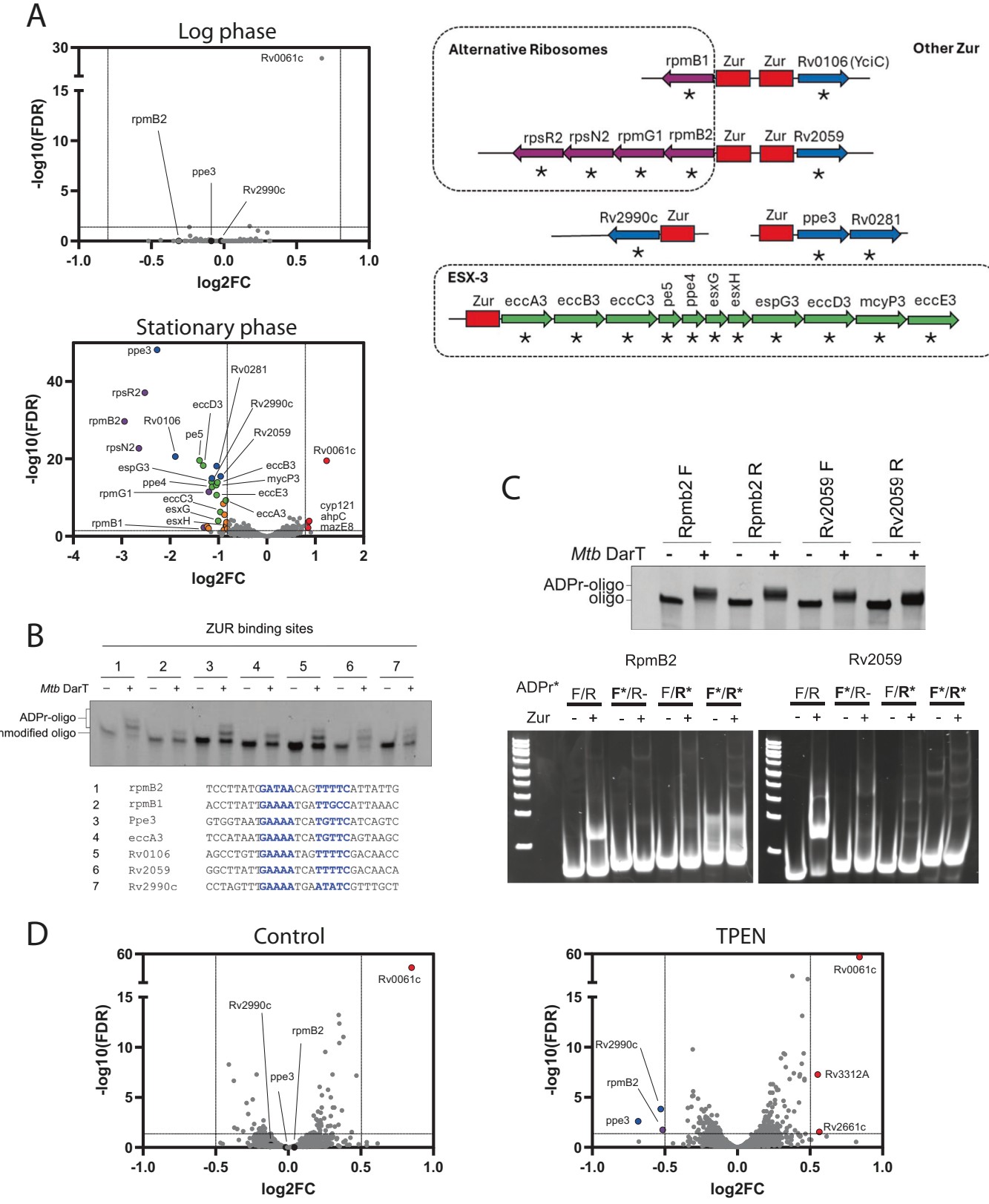

Figure 4.  Regulation of stationary phase gene expression by DarT-dependent ADP-ribosylation of DNA in *M. tuberculosis.*

(A) RNA-Seq of wild-type *M. tuberculosis* GC1237 versus *M. tuberculosis ΔdarTdarG* in logarithmic phase growth at 6 days (upper panel) and stationary phase growth at 20 days (lower panel). Genes with significantly altered transcription between wild-type and ΔdarTdarG-knockout strain (FDR <0.05) and a log2 fold change (log2FC) of >0.8 (*n* = 4) are coloured, and functional category is indicated in the colour-coordinated genomic organisation diagram (right panel, *significantly altered transcription, Zur-binding sites in red). (B) DarT-dependent ADP-ribosylation of Zur-binding site DNA sequences from the promoters of genes differentially regulated in stationary phase *M. tuberculosis ΔdarTdarG.* Zur-binding motifs are highlighted in blue. (C) DarT-dependent ADP-ribosylation of either DNA strand in Zur-binding sites from the divergent promoter between *rpmB2* and Rv2059 inhibited Zur binding to DNA. (D) RNA-Seq of wild-type *M. tuberculosis* GC1237 versus *M. tuberculosis ΔdarTdarG* in zinc-replete media (left panel) and under zinc deficiency by exposure to 1.2 µM TPEN (right panel). Genes with significantly altered transcription between wild-type and ΔdarTdarG-knockout strain (FDR <0.05) and a log2FC of >0.5 are coloured. *n* = 4. Source data are available online for this figure.

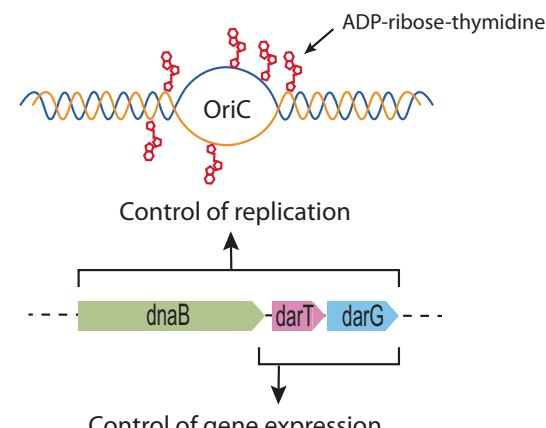

Control of replication

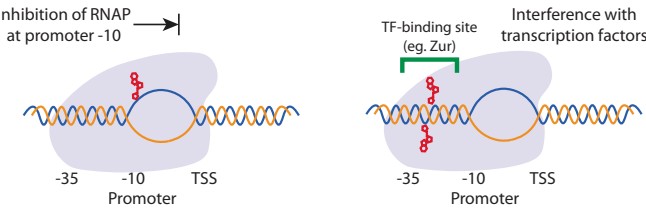

Control of gene expression

**Figure 5.   Coordination of DNA replication and transcription by DarT-dependent ADP-ribosylation of thymidine in *M. tuberculosis* complex bacteria.**

DarT ADP-ribosyltransferase and DarG ribosylglycohydrolase (transcriptionally linked to the replicative helicase DnaB) reversibly modify thymidine at the chromosome origin of replication (OriC) to regulate DNA replication. In coordination, ADP-ribosylation regulates gene expression by modification of thymidine at the promoter to inhibit escape of RNAP from the promoter or alter the binding of transcription factors.

broth containing 25 µg/ml kanamycin and 50 µg/ml hygromycin. The bacteria were adjusted to OD$_{600}$ nm 0.05 in roller bottles containing 7H9 broth, and cultured for 48 h in the presence or absence of 200 ng/ml anhydrotetracycline (ATC) to induce *darG* silencing. Genomic DNA was extracted. Cultures were centrifuged and resuspended in TE pH 8.0. An equal volume of methanol:-chloroform (2:1) was added, and cultures were rocked for 5 min by hand. The bacilli were centrifuged and the pellet dried, then resuspended in TE pH 8.0 and distributed into Lysing Matrix B tubes (MP Biomedicals). An equal volume of phenol:chloroform:i-soamyl alcohol (25:24:1) was added, and the bacilli were disrupted using a FastPrep homogeniser. The mixture was centrifuged to pellet the debris and to separate the phases. The upper aqueous

phase was subjected to a further extraction with phenol:chlor-oform:isoamyl alcohol, and two extractions with chloroform. gDNA was precipitated with 0.1 volumes of 3 M sodium acetate pH 5.2, and 1 volume of propan-2-ol, pelleted by centrifugation, washed with 70% ethanol, resuspended in TE pH 8.0 and frozen at −20 °C. DNA concentration was determined using the QuantiFluor dsDNA system (Promega). 10 µg aliquots of gDNA were digested in a total volume of 250 µL with 20 U of Alu1 enzyme (NEB) and 1 µg RNAse (DNase-free; Roche) in 1× CutSmart buffer, overnight at 37 °C. Digestion was verified by electrophoresis on a 1.5% TAE-agarose gel stained with 1× SYBRSafe. Fragmented DNA was purified using the Genomic DNA Clean and Concentrator Kit – 10 (Zymed), following the manufacturer's protocol for DNA fragments.

ARPr-immunoprecipitation was performed essentially as previously described (Schuller et al, 2021). 500 ng of Alu1-digested gDNA was retained as the input sample, and 4.5 µg of Alu1-digested gDNA was placed in a low bind-Eppenforf tube with 500 µL Immunoprecipitation buffer (10 mM NaPO$_4$ pH 7.0, 140 mM NaCl, 0.05% Triton X-100), and 4.5 µg of rabbit anti-poly/mono-ADP-ribose antibody (E6F6A, Cell Signaling Technology). The tubes were rotated for 4 h at 4 °C, and 30 µl of Protein A Dynabeads (Invitrogen) equilibrated in IP buffer were added and the tubes rotated overnight at 4 °C. The Dynabeads were then washed three times in cold IP buffer, using a Dynamag magnet. The beads were resuspended in 400 µL digestion buffer (10 mM TRIS pH 8.0, 100 mM EDTA, 0.5% SDS, 50 mM NaCl, 100 µg proteinase K per tube) and digested for 4 h at 55 °C. The beads were removed by magnetic separation, and the released DNA fragments were purified from the supernatant using Zymo ChIP DNA clean and concentrate columns (Zymed), according to the manufacturer's instructions. Samples were eluted in 20 µL TE buffer containing 0.1 mM EDTA, and 1ul used to quantify DNA yield. Three pulldowns each of uninduced and ATC-induced (*darG*-silenced) DNA fragments demonstrated sufficient specific ADPr-enrichment and yield (28–46 ng per reaction) to allow NGS sequencing. Illumina TruSeq Pair-Ended Sequencing was performed by Azenta (Genewiz). Pilot experiments indicated that recovery from pull-downs carried out using an isotype-matched antibody control would not yield sufficient material for sequencing.

## ADPr-Seq analysis

The raw sequencing reads were pre-processed with fastp (version 0.23.4) (Chen, 2023) to remove sequencing adaptors and low-quality bases. The reads were then aligned to the reference genome of *M. bovis* subsp. *bovis* strain AF2122/97 (Malone et al, 2017)

using bwa-mem2 (version 2.2.1) (Vasimuddin et al, 2019) with default settings. Misaligned and multimapping reads were filtered out by excluding reads with MAPQ scores below 30 and retaining only primary alignments (–F 256) and properly paired reads (–f 2) using samtools (v1.18) (Danecek et al, 2021). A custom R script was used to count paired reads that mapped to both ends of each DNA fragment generated by the Alu1 restriction enzyme across the genome. The quantified fragments were normalised to counts per million (CPM) before being used in the differential analysis. We analysed the relative abundance of ADP-ribosylated DNA fragments using a negative binomial generalised linear regression of read counts to estimate the relative enrichment of ADPr fragments in the induced DarG-knockdown mycobacteria versus uninduced mycobacteria. Relative *Thymine-ADP-ribose enrichment* for DNA fragments in *darG*-knockdown mycobacteria was defined as *log((ADP-ribose-DNA enrichment of ATC-induced BCG darG-sgRNA relative to unenriched input) / (ADP-ribose-DNA enrichment of uninduced BCG darG-sgRNA relative to unenriched input))* and plotted against genome position.

## RNA-Seq

For RNA-Seq comparisons between wild-type *M. tuberculosis* GC1237 and *M. tuberculosis ΔdarTdarG*, bacteria were cultured in 7H9-ADC broth for 6 days (logarithmic phase) or 20 days (stationary phase) with shaking at 37 °C. For comparison of wild-type and *ΔdarTdarG* strains with and without the zinc chelator *N,N,N′,N′*-Tetrakis(2-pyridylmethyl)ethylenediamine (TPEN), bacteria were cultured in Sauton's broth (containing 1 μM $ZnSO_4$) for 6 days before addition of TPEN to 0.6 μM (concentration non-inhibitory to growth) or DMSO control for 24 h. Four replicates were performed for all strain/treatment combinations. Cultures were crash-cooled with frozen PBS and centrifuged at 4 °C for 10 min at $4100 \times g$. Cell pellets were resuspended in 1.2 ml Trizol and RNA extracted as per the manufacturer's instructions with the final RNA pellet being dissolved in 40 μl water. The RNA was DNase (Qiagen) treated for 10 min at 30 °C followed by a second on-column DNase treatment using the Zymo RNA Clean and Concentrate columns and finally eluted in water. RNA concentration and integrity were assessed by Nanodrop and Agilent RNA Nano 6000 chips using an Agilent Bioanalyser. RNA was depleted of ribosomal RNA, fragmented and random primed for first and second strand cDNA synthesis. cDNA was end-repaired, 5' phosphorylated and dA-tailed before adaptor ligation, PCR enrichment and sequencing on Illumina NovaSeq (Genewiz). Reads were trimmed and aligned to the *M. tuberculosis* H37Rv genome (Accession: NC_000962). After the extraction of gene hit counts, DESeq2, was used to compare gene expression between groups.

## Viability and microscopy assays to show reversible growth inhibition by DarT

*M. bovis* BCG bacilli carrying CRISPRi *darG2*-knockdown plasmid pIJR965-DarG2 were grown to late log phase in 7H9-10% ADC supplemented with 25 μg/ml kanamycin. Cultures were filtered through a 70-μM mesh, and the $OD_{600}$ nm adjusted to 0.05. Bacteria were seeded into a 96-well plate, and a twofold dilution series of anyhydrotetracycline was added to induce *darG* silencing in a graded manner. Plates were sealed and incubated for 7 days at

37 °C. After 7 days, growth was assessed by measuring $OD_{600}$ nm using a plate reader, and wells containing the minimum concentration of ATC that prevented growth were determined (typically 2–8 ng/ml). Bacilli were harvested by centrifugation, resuspended in fresh medium, and re-seeded in 96-well plates with or without ATC at the concentration used previously. Growth was assessed by measuring $OD_{600}$ nm after a further 7 days. Data were normalised by dividing the value by the mean of untreated control values. For microscopy experiments, minimally *darG*-silenced bacilli were harvested after 7 days of treatment and loaded into a CellAsic ONIX Microfluidic Platform using a plate for bacterial cells. Bacilli were visualised using the 561 nm laser line and transmitted light detector on a Nikon A1M Confocal Microscope (on a Ti-E microscope) in a heated environmental chamber using a CFI Plan Apochromat Lambda D 40x air lens (numerical aperture 0.95) and the Nikon Perfect Focus System to maintain focus. Time-lapse images were taken each hour for 4 days, while the microfluidic pump maintained of medium with or without ATC. Images were analysed using NIS-Elements software.

## Recombinant Zur expression

An expression construct for N-terminal His-tagged Zur with a TEV cleavage site was synthesised by inserting the Zur gene into the pET151/D-TOPO plasmid. Plasmid construction, synthesis and sequencing were carried out by Invitrogen (Thermo Fisher). For protein expression, BL21(DE3) pLysS bacilli were transformed by heat shock and recovered on LB plates containing 100 μg/ml ampicillin. Overnight starter cultures were prepared from freshly transformed colonies, and 500 μL culture was used to inoculate 50 ml of fresh pre-warmed LB-Amp medium. Cultures were grown shaking at 250 rpm at 37 °C to an $OD_{600}$ nm of 0.5, and protein expression induced using 0.4 mM IPTG for 4 h. Cultures were harvested by centrifugation ($4500 \times g$) for 25 min at room temperature, and the pellets frozen at −20 °C. The pellet was thawed and lysed using B-PER™ Complete Bacterial Protein Extraction Reagent (Thermo Fisher) supplemented with EDTA-free protease inhibitors (Pierce), for 45 min at room temperature. The lysate was clarified by centrifugation, and applied to a His GraviTrap TALON column (Cytiva) that had been equilibrated with equilibration buffer (50 mM sodium phosphate, 300 mM NaCl, pH 7.4). The column washed with equilibration buffer containing 5 mm imidazole, and the his-tagged protein eluted using equilibration buffer containing 150 mM imidazole. The his-tag was removed by digestion overnight at 4 °C using his-tagged TEV protease (Sigma Aldrich). The reaction was buffer exchanged into protein storage buffer (20 mM Tris pH 7.5, 100 mM NaCl, 10% glycerol, 50 mM L-arginine). The protein was further purified by reverse IMAC using Ni-NTA beads to remove digested his-tags, his-TEV, and undigested his-Zur protein. Aliquots of purified Zur were frozen at −70 °C.

## Recombinant DarT expression

Expression and purification of *M. tuberculosis* DarT for the in vitro studies was performed essentially as previously described (Schuller et al, 2021). The pBAD33 vector encoding for *M. tuberculosis* DarT was transformed into *E. coli* 'DH5α-macro' cells. Cells were grown at 37 °C in LB medium (Miller) supplemented with 25 μg/mL chloramphenicol and 0.8% (*w/w*) glucose to an $OD_{600}$ nm of 0.8.

Cells were then pelleted by centrifugation at 4000×g for 15 min at RT and resuspended in fresh LB media containing 25 µg/ml chloramphenicol and 0.8% (w/w) arabinose to induce protein expression. After 2.0 h at 37 °C, cells were harvested by centrifugation (4000×g, 15 min) and resuspended in lysis buffer (50 mM TRIS-Cl (pH 8.0), 500 mM NaCl, 5% glycerol, 20 mM imidazole, 0.5 mM TCEP) and stored at −20 °C until purification. Cells were lysed using BugBuster (Novagen) following the manufacturer's instructions after adding cOmplete EDTA-free protease inhibitors (Roche) and benzonase (Novagen). *M. tuberculosis* DarT protein was purified by IMAC using Ni-Sepharose resin (GE Healthcare) which was followed by buffer exchange by repeated filter concentration in storage buffer containing 50 mM HEPES (pH 7.4), 300 mM NaCl, 5% glycerol, 0.5 mM TCEP. The purified protein was characterised by SDS-PAGE, then flash frozen in liquid nitrogen and stored at −80 °C until required. Protein concentrations were determined by measuring absorption of the sample at 280 nm with the NanoDrop1000 (Thermo Scientific).

## In vitro ADP-ribosylation of DNA

ADP-ribosylation reactions were performed in ADP-ribosylation buffer (50 mM Tris-Cl pH 8.0, 50 mM NaCl) supplemented with 5 mM EDTA at 37 °C for 30 min. 1 µM *M. tuberculosis* DarT protein was incubated with oligonucleotides at a concentration of 3 µM and β NAD$^+$ in excess (500 µM). Reaction products were analysed by separation on denaturing polyacrylamide gels run in TBE buffer, loading 0.02 nmol oligonucleotide with urea loading dye (10 mM TRIS pH 8.0, 10 mM EDTA, 4 M urea), and by following visualisation under UV light (340 nm) after ethidium bromide-staining. Representative results of two independently performed experiments are shown.

## Large-scale preparation of ADPr DNA for in vitro transcription and EMSA

2500 nmole of *rrnAP3* promoter oligonucleotides (Table EV2) were ADP-ribosylated with *M. tubercuolosis* DarT by incubation at 37 °C O/N in presence of 500 µM β-NAD$^+$ in ADP-ribosylation buffer (50 mM Tris-Cl pH 8.0, 50 mM NaCl) supplemented with 5 mM EDTA. The ADP-ribosylated oligonucleotides were gel-purified on denaturing polyacrylamide gel run in TBE buffer, and recovered by excising from the gel and extracting from the gel pieces by diffusion with several elution steps in TE buffer. Gel residuals were removed by centrifugation of the eluates through 0.22-µm filter units (GE Healthcare). Finally, eluates were desalted using Sephadex G-25 in PD-10 Desalting Columns (GE Healthcare) equilibrated in nuclease-free water and concentrated to the desired concentrations.

## Electromobility shift Assay (EMSA) of Zur/DNA interaction

The oligonucleotides used for the EMSA assay are detailed in Table EV2. Briefly, 60mer oligonucleotides were ADP-ribosylated using DarT enzyme as detailed above using either the DarT toxin or a mock reaction. Following labelling, the absorptions at 260 nm were calculated and primers annealed in a 1:1 ratio at a final concentration of 5 µM in Annealing Buffer (10 mM Tris pH 7.5, 50 mM NaCl) by heating for 7 min at 95 °C before allowing to cool

to room temperature. Primers were stored at −20 °C until further use. EMSA reactions (20 µL total volume) were carried out in EMSA buffer (20 mM Tris HCl pH 7.5, 1 mM L-arginine, 10% glycerol) and contained 10 picomoles of annealed oligonucleotides, 400 µM EDTA, 500 µM ZnCl2, and 2.5 µg (167 pmoles) of purified Zur protein. After incubating for 30 min at room temperature, the whole reaction run at 150 V on a non-denaturing 6% polyacrylamide gel (acrylamide/bis-acrylamide 37.5:1) containing 2.5% glycerol in 0.25× Tris-borate buffer pH 8.0. The gel was counterstained with 1× SYBR Gold and visualised using a Bio-1000F gel imager (Microtek).

## In vitro transcription

*M. tuberculosis* RNAP holoenzyme, RNAP core enzyme, were purified following the protocol previously described (Banerjee et al, 2014). Plasmids pETDuet *rpoB-rpoC* and pAcYc Duet-*rpoA-sigA* or pAcYc Duet-*rpoA-rpoz* were co-transformed into *E. coli* BL21 (DE3) cells for purifying *M. tuberculosis* RNAP. The cells were grown in 1 L 2XYT (16 g Tryptone, 10 g Yeast Extract, 5 g NaCl, pH 7.0) at 37 °C until OD (at 595 nm) reached 0.4. Then 0.5 mM IPTG was added to the culture, and the culture containing the cells was allowed to grow for another 12 h at 16 °C to induce protein production in the cells. Then, the cells were harvested by centrifugation at 5000× g for 10 min at 4 °C. After harvesting the cells, *M. tuberculosis* RNAP holo was purified using Ni-NTA affinity chromatography and ion exchange chromatography (Mono Q) consecutively.

DNA substrates for in vitro transcription were: wild-type *rrnAP3* promoter from −59 to +47; *rrnAP3/ADPr* which had ADP-ribose modification of the non-template (NT) strand; *rrnAP3+* carrying an g to t base change at +23 to introduce several DarT modification sites to the template strand of the transcribed region; and *rrnAP3 + /ADPr* which had ADP-ribose modification of the template strand (Table EV2). 20 nM DNA substrate was incubated with the RNAP holoenzyme in 10 µl transcription reaction buffer [45 mM Tris-Cl (pH 8), 5 mM MgCl$_2$, 70 mM KCl, 1 mM DTT, 10% glycerol, 1.5 mM MnCl$_2$] at 37 °C for 20 min. 1 µl heparin (0.5 mg/ml) was added to the reaction to inhibit the non-specific RNAP-DNA complex. RNA synthesis was initiated by adding 1 µl NTP mix (final concentration: 125 mM of ATP, GTP, CTP and 50 mM of UTP containing 0.4 mCi a32$^P$-UTP). Following incubation at 37 °C for 30 min, the reactions were terminated by 5 µl FLB dye (80% formamide, 10 mM EDTA, 0.01% bromophenol Blue, 0.01% xylene cyanol). Products were heated at 95 °C for 5 min, chilled on ice, and resolved by 12% Urea-PAGE, scanned by storage Phosphor Imager scanner (Amersham Typhoon).

## Electrophoretic mobility shift assay (EMSA) of RNA polymerase interaction with *rrnAP3* promoter

In total, 50 nM of *rrnAP3* wild-type DNA fragments were mixed with 0, 100, 200 nM *M. tuberculosis* RNAP-SigA holo in 10 µl transcription reaction buffer and incubated at 37 °C for 15 min to form RNAP-DNA complexes. 200 nM RNAP core enzyme was used in a separate reaction as a control. Heparin (0.5 µg) was added to the samples before resolving on 6% PAGE in 1× TBE buffer (89 mM Tris base, 89 mM boric acid, 2 mM EDTA, pH 9) under

100 V at 4 °C for 1 h. The gels were stained by SYBR gold (Thermo Scientific) for 30 min and scanned by Fluorescence Scanner (Amersham Typhoon) at DyLight 488 channel.

Separate EMSA assays were performed using the same protocol but with the *rrnAP3*/ADPr DNA template which carries ADP-ribosylation of the non-template strand as used in the IVT assay.

## Statistical analysis

Bacterial cell lengths were compared between treatments by one-way ANOVA. ADPr-Seq and RNA-Seq analyses are described above. The transcriptional activity of genes in different ADPr modification states was compared by one-way ANOVA with Holm–Sidak multiple comparisons test applied to log2 fold change expression in *darG*-knockdown bacteria compared to uninduced controls.

## Data availability

The datasets in this study are available in the following databases: RNA-Seq: Gene Expression Omnibus GSE283124 for *M. tuberculosis* wild-type vs Δ*darTdarG* in log and stationary phase GEO Accession viewer. GSE283122 for *M. tuberculosis* wild-type vs Δ*darTdarG* ± TPEN GEO Accession viewer. ADPR-Seq: Gene Expression Omnibus GSE283123. GEO Accession viewer. The numerical data used to generate the graphs in this study is available in the Source Data. All other supporting data are available from the corresponding author upon reasonable request.

The source data of this paper are collected in the following database record: biostudies:S-SCDT-10_1038-S44318-025-00451-y.

## Peer review information

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

## Acknowledgements

This work was supported by the Biotechnology and Biological Sciences Research Council [grant number BB/W016613/1]. The work in IA laboratory is also supported by the Wellcome Trust (grants 223107 and 302632), the Guy Newton Translation Fund (grant GN05 18) and the CRUK (grant C35050/A22284). MS is additionally supported by the Goodger and Schorstein Scholarships Trust Fund (Source of funds BC076). The authors thank David Jones for SEM, Linda McLatchie for assistance with protein purification, Hannah Stewart for microbiology and the University of Surrey Bioimaging Facility.

## Author contributions

**Rachel E Butler**: Funding acquisition; Validation; Investigation; Methodology; Writing—original draft; Writing—review and editing. **Marion Schuller**: Formal analysis; Validation; Investigation; Methodology; Writing—original draft. **Ritu Jaiswal**: Validation; Investigation. **Jayanta Mukhopadhyay**: Formal analysis; Supervision; Methodology; Writing—original draft. **Jim Barber**: Investigation. **Suzie Hingley-Wilson**: Funding acquisition; Methodology. **Emily Wasson**: Data curation; Formal analysis. **Alex Couto Alves**: Formal analysis; Supervision; Investigation. **Ivan Ahel**: Conceptualisation; Supervision; Funding acquisition; Writing—original draft. **Graham R Stewart**: Conceptualisation; Funding acquisition; Investigation; Methodology; Writing—original draft; Writing—review and editing.

Source data underlying figure panels in this paper may have individual authorship assigned. Where available, figure panel/source data authorship is listed in the following database record: biostudies:S-SCDT-10_1038-S44318-025-00451-y.

## Disclosure and competing interests statement

The authors declare no competing interests.

# Expanded View Figures

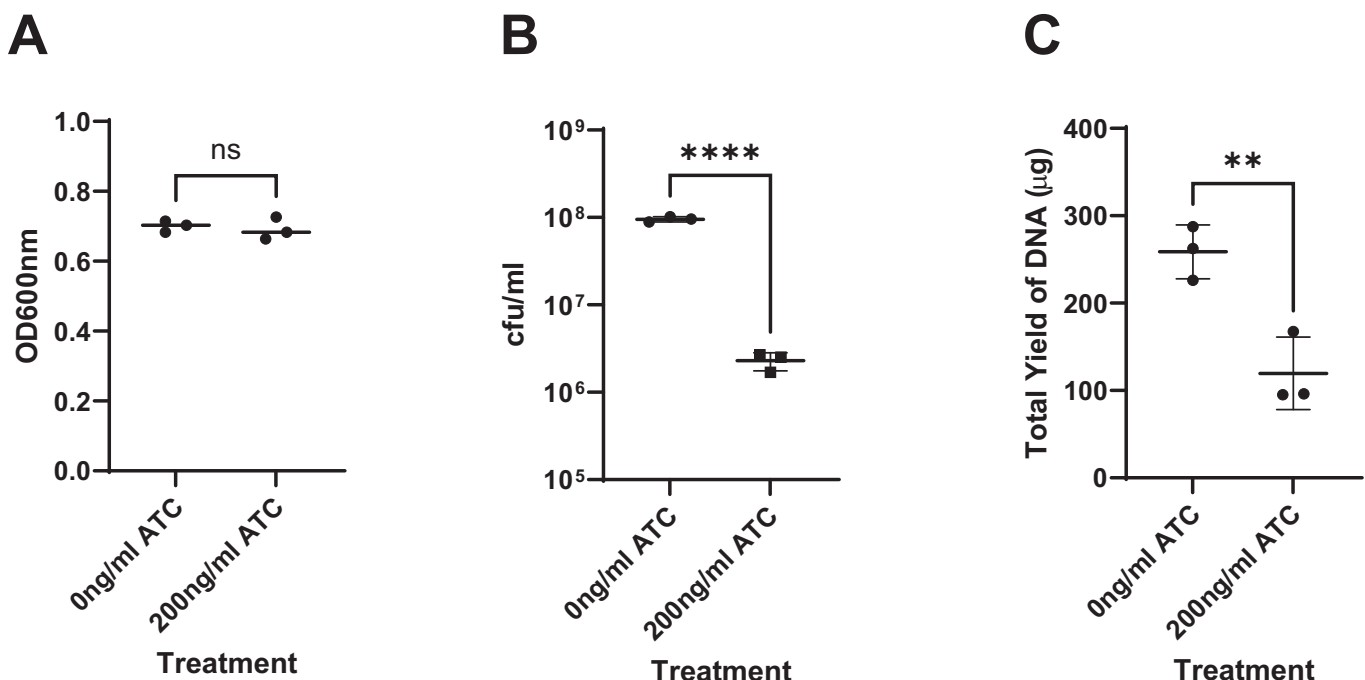

**Figure EV1.  *In cellulo* ADP-ribosylation of gDNA inhibits DNA synthesis.**

In cellulo ADP-ribosylation of gDNA in *M. bovis* BCG was induced in late log phase by CRISPRi of the antitoxin *darG* with 200 ng/ml ATC (or 0 ng/ml ATC uninduced control). After 24 h, the cultures were diluted to stimulate replication, and the ATC concentration replenished as indicated. After 3 days, optical density ($OD_{600}$ nm; **A**), and colony forming units (cfu; **B**)) were assessed. Genomic DNA (**C**) was isolated from the cultures by phenol:chloroform:isoamyl extraction and quantified as detailed in "Methods" section. Data are mean $+/-$ SD for $n = 3$ replicate cultures; n.s = not significant, **$P = 0.0096$. ****$P = 0.00002$ by unpaired two-tailed *T* test. Source data are available online for this figure.

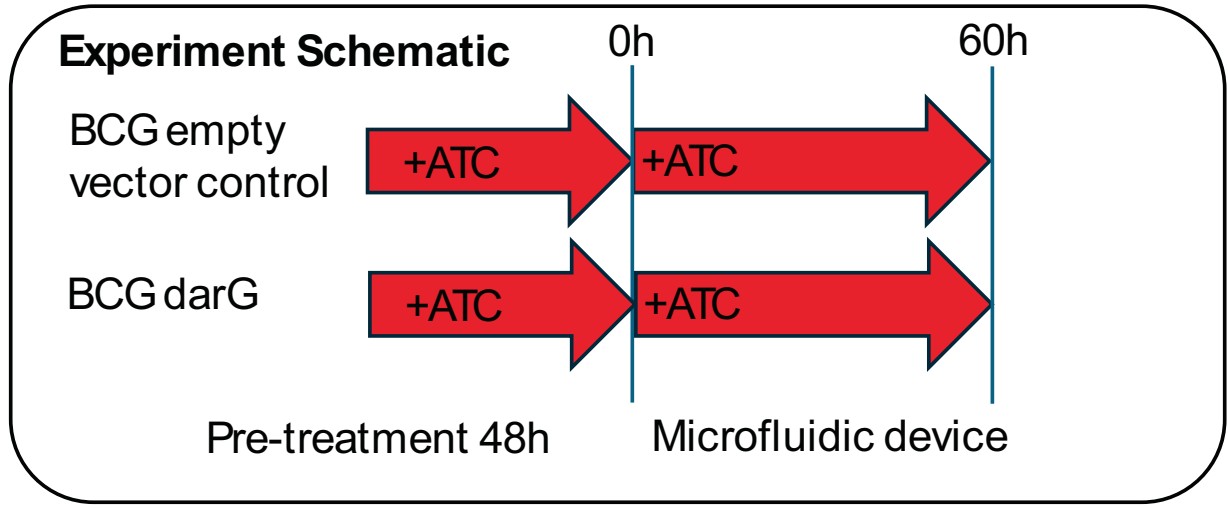

## BCG empty vector control (Movie EV1)

0h        29h        59h

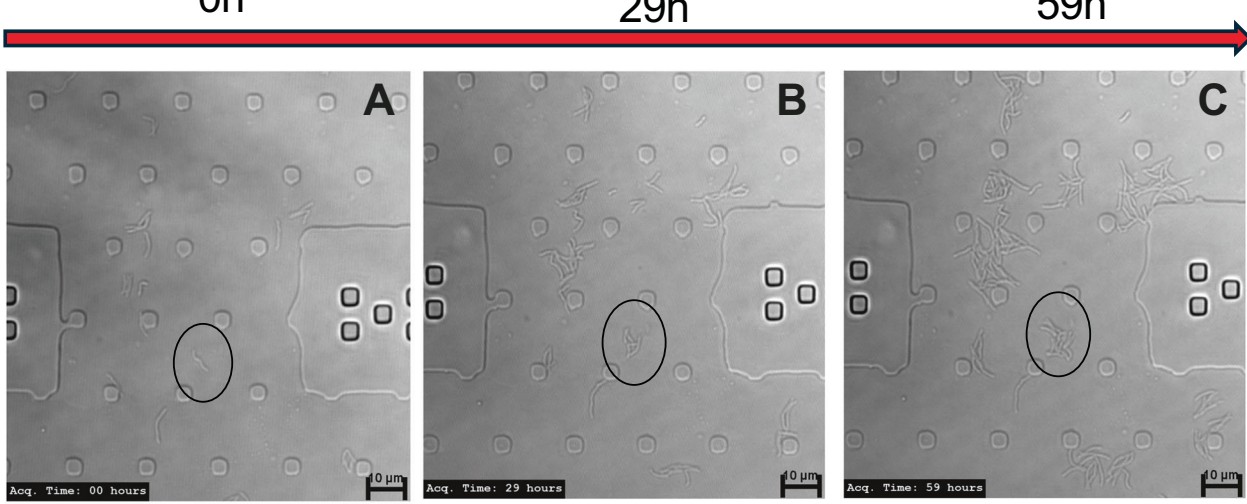

## BCG *darG* sgRNA (Movie EV2)

0h        29h        59h

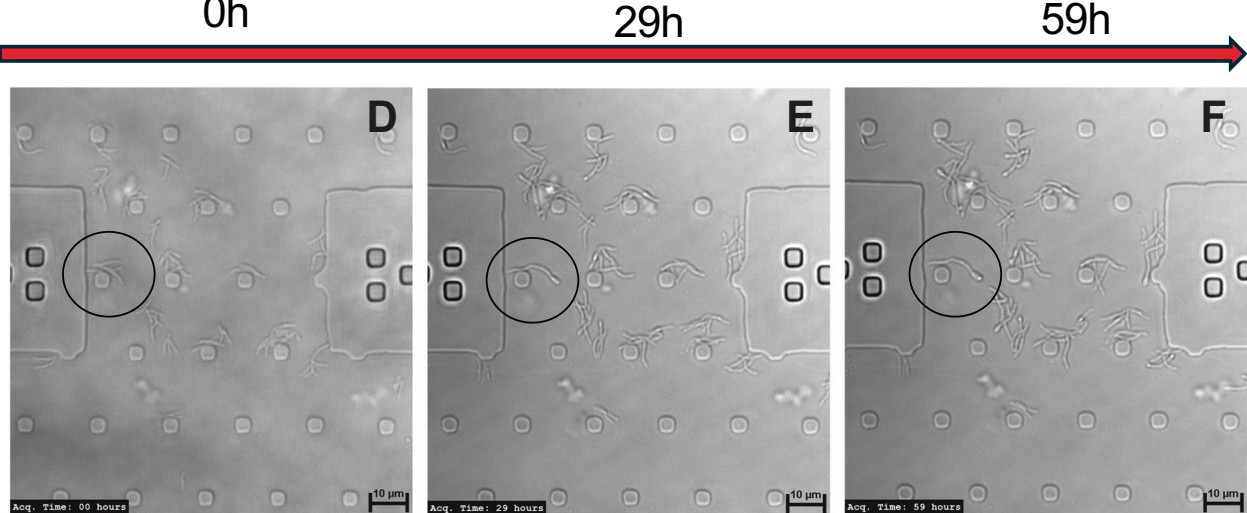

**Figure EV2. Time-lapse stills from Movies EV1 and EV2.**

An experiment schematic is included in the panel. BCG darG-sgRNA and BCG carrying an empty vector were pretreated for 48 h with 200 ng/ml ATC, then loaded into a microfluidic device. The bacteria were imaged every hour for 59 h, with continued ATC treatment. BCG-vector control bacilli divide over 59 h (**A–C**), whereas BCG darG-sgRNA stop dividing and elongate (**D–F**). An example is circled in each treatment. Source data are available online for this figure.

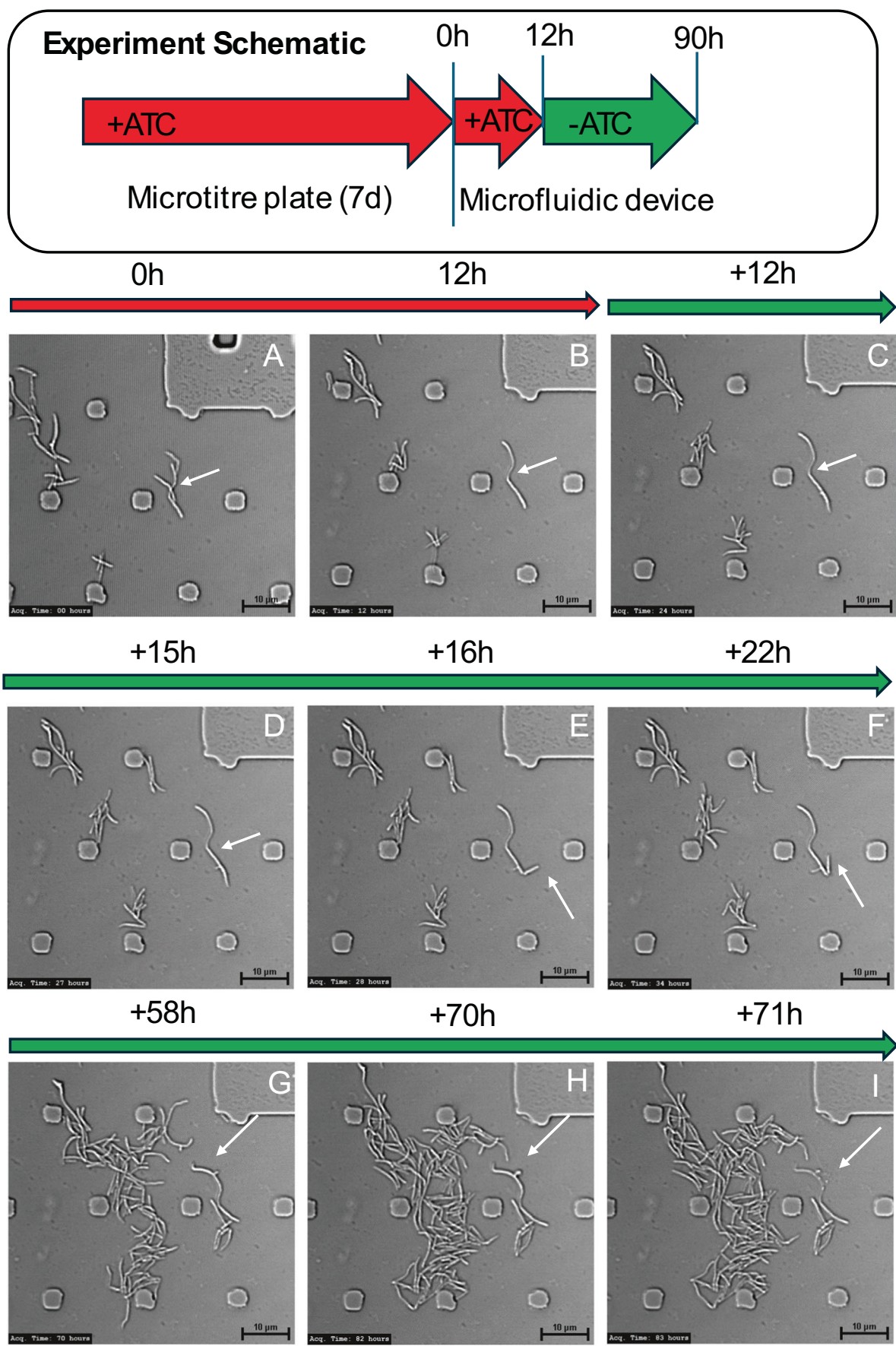

**Figure EV3.** **Time-lapse stills from Fig. 1D and Movie EV3.**

An experiment schematic is included in the panel. BCG darG-sgRNA were minimally inhibited with ATC in a microtitre plate for 7 days, and loaded into a microfluidic device (**A**). After 12 h, ATC was washed out of the device (**B**). The bacterium marked with a white arrow was followed over time (**C**). After 16 h in fresh medium, division occurs at one pole (**D–F**). After 58 in fresh medium, branching occurs at the other pole (**G**) but is followed by swelling (**H**) and necrosis (**I**).

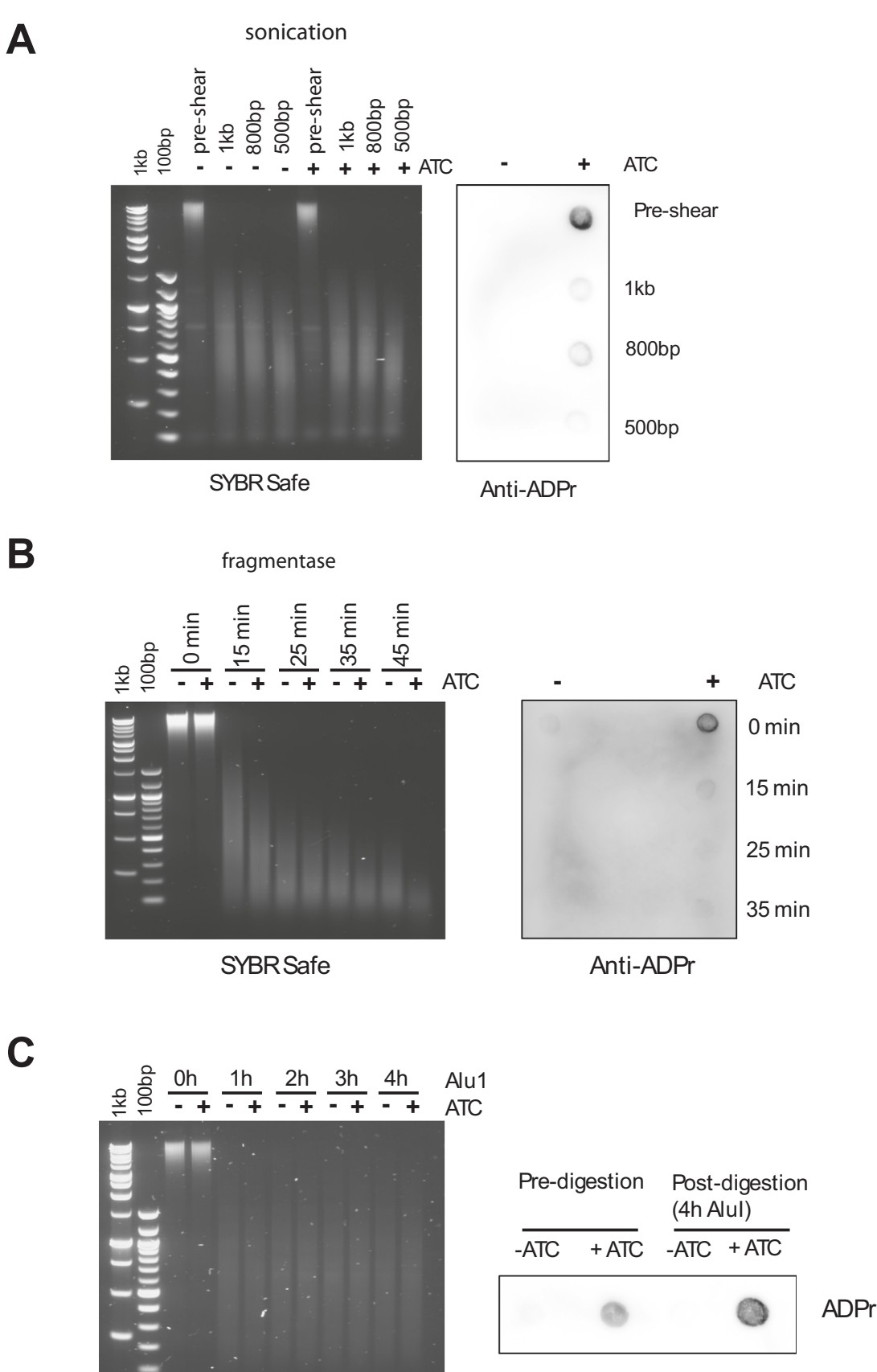

◀ **Figure EV4. Survival of the E6F6A ADPr epitope after gDNA fragmentation.**

In cellulo ADP-ribosylation of gDNA was induced in BCG by CRISPRi knockdown of the antitoxin *darG* with 200 ng/ml ATC (or 0 ng/ml ATC control) for 48 h. gDNA was extracted and fragmented by (**A**) sonication (S220 Focussed Ultrasonicator, Covaris); (**B**) NEBNext Fragmentase Enzymes (FR); (**C**) AluI digestion. Fragmentation was analysed by agarose gel electrophoresis (left panels). Equal quantities of DNA were bound to a Zetaprobe membrane and epitope stability assessed by dot-blot using the E6F6A antibody against ADP-ribose. The ADPr-Seq technique requires the generation of gDNA fragments that are affinity purified using the anti-ADPr antibody E6F6A, which are then sequenced by NGS. This approach required the ADPr epitope to remain on the DNA following fragmentation. We tried three approaches to generating gDNA fragments. Typically sonication is used to generate 500 bp fragments for NGS studies, however this led to the loss of the ADPr epitope (**A**). Random fragmentation using NEBNext dsDNA Fragmentase also led to loss of the ADPr epitope, and unequal fragment sizes were generated between control gDNA (-ATC) and ADPr-gDNA (+ATC) (**B**). Digestion with AluI maintained the epitope post fragmentation, and resulted in a similar fragmentation pattern in control gDNA (-ATC) and ADPr-gDNA (+ATC) (**C**).

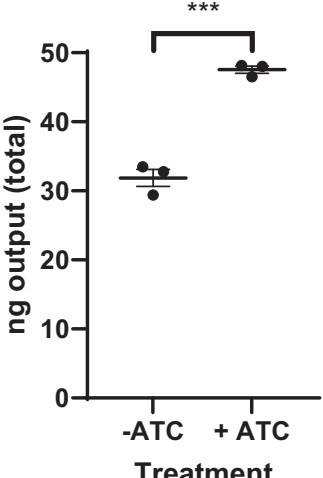

**Figure EV5.  Yield of DNA fragments recovered by immunoprecipitation with anti-ADPr antibody.**

DNA from *darG*-silenced (+ATC) and uninduced control BCG (-ATC) was fragmented with AluI and immunoprecipitated using the E6F6A anti-ADPr antibody. The yield of DNA from each precipitation was measured using the dsDNA Quantifluor system (Promega). Error bars are mean +/− SEM, ***$P = 0.0003$ by unpaired T test.

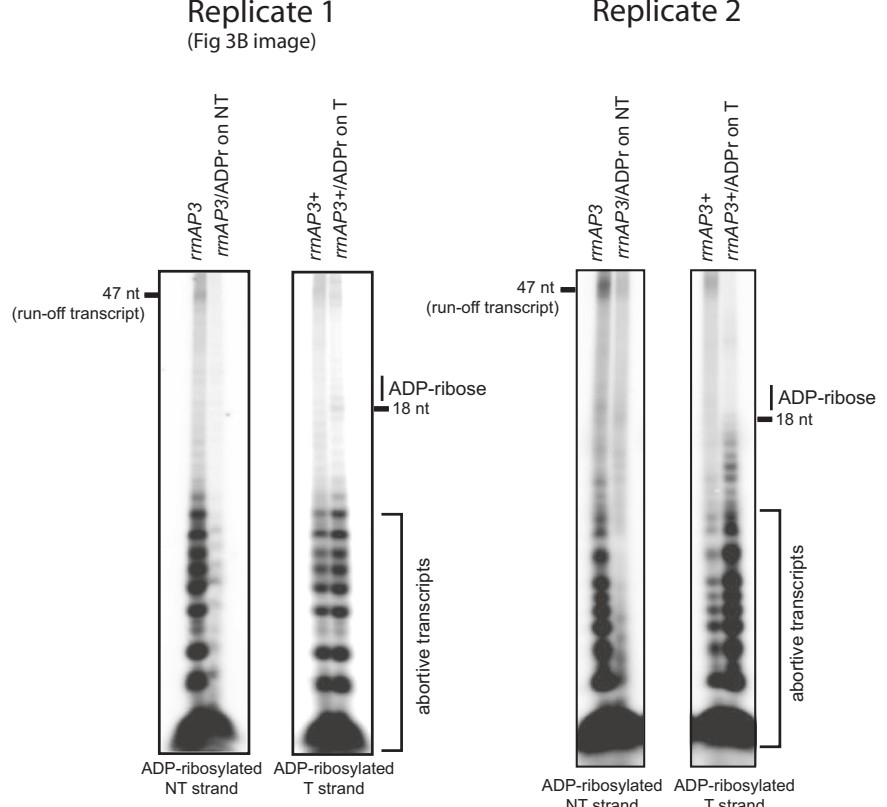

*in vitro* transcription

**Figure EV6.   Replicate in vitro transcription assay showing effects of ADP-ribosylation of ssDNA on transcription by *M. tuberculosis* RNA polymerase.**

IVT was performed with the *M. tuberculosis* RNAP holoenzyme on *rrnAP3*/ADPr template (ADP-ribose on non-template NT strand) and *rrnAP3 + /*ADPr template (ADP-ribose on template T strand) alongside substrates carrying no ADP-ribose modifications. This showed that ADP-ribosylation of the NT strand of *rrnAP3* inhibited the formation of both abortive transcripts as well as run-off transcripts, whereas ADP-ribosylation on the template strand of *rrnAP3+* did not inhibit transcription although there was some indication for formation of stalled elongation complex near the introduced ADP-ribosylation site. For direct comparison, Fig. 3B image (replicate 1) is shown alongside replicate 2.

