## [Peer Review File · The EMBO Journal]

Control of replication and gene expression by ADP-ribosylation of DNA in *Mycobacterium tuberculosis*

Rachel Butler, Marion Schuller, Ritu Jaiswal, Jayanta mukhopadhyay, Jim Barber, Suzie Hingley-Wilson, Emily Wasson, Alex Couto Alves, Ivan Ahel, and Graham Stewart

Corresponding author(s): Graham Stewart (g.stewart@surrey.ac.uk) , Ivan Ahel (ivan.ahel@path.ox.ac.uk)

Review Timeline:

Submission Date:	30th Jan 25
Editorial Decision:	21st Feb 25
Revision Received:	26th Mar 25
Editorial Decision:	2nd Apr 25
Revision Received:	10th Apr 25
Accepted:	22nd Apr 25

Editor: Hartmut Vodermaier

Transaction Report:

This manuscript was transferred to The EMBO JOURNAL following peer review at another journal.

Dr. Graham Stewart
University of Surrey
United Kingdom

21st Feb 2025

Re: EMBOJ-2025-120334-T
Control of replication and gene expression by ADP-ribosylation of DNA in *Mycobacterium tuberculosis*

Dear Dr. Stewart,

Thank you for transferring your previously reviewed manuscript, together with referee reports and revision plans, to The EMBO Journal; and apologies for the delay in getting back to you with a definitive response. As already communicated in our transfer offer, we would be interested in publishing your study without major experimental extensions, provided that specific issues raised by the reviewers can be adequately addressed. I have now had the chance to consider your responses to the referees' points in detail, and found them overall quite promising (see specific comments below). In conclusion, I am therefore happy to invite you to revise the manuscript based on your tentative point-by-point response and according to EMBO Journal editorial policies, as follows:

Scientific points:

- For referee 1, please incorporate the clarifications and presentational amendments, including the additional data/images, as outlined in your revision plan. I agree that Figure 1E may best be removed at this point.
- For Figure 3, I think that the labeling has to be improved (e.g., the legend says "lane 5, ... rrnA1-mut2 ... on the non-template strand" but "lane 6, rrnA1-mut2 ... of the template strand"?), although regarding the annotation of the three separate panels (and cf. ref. 3's related concern). This all makes it currently somewhat difficult to appreciate your response to major point 2 of referee 2, but do I understand correctly that you would propose to only retain lanes 1-4 & conclusions regarding the non-template strand in the final version? Please note that the additional replicates you mention for the template strand pausing/stalling, and/or quantifications, could easily be included as Expanded View or Appendix figures, with tempered conclusions if necessary.
- Regarding the use of the contentious term "epigenetic", I am afraid I have to concur with referee 2 (and also referee 1) that it should only be used in the true sense of the meaning - to indicate traits being transmitted to daughter cells by means beside ("epi") the sequence-hardwired genetic inheritance (e.g., through passing on of a chromatin modification, of an RNA species, a prion protein...); but not as a collective term for every post-translational modification that reversibly affects gene expression. I would therefore appreciate removing the term from title, abstract, keywords and throughout the text, and instead switching to a more concrete terminology. I like the alternative title you suggest in response to referee 3, "Control of replication and gene expression by ADP-ribosylation of DNA in *Mycobacterium tuberculosis*."
- For referee 3, I appreciate that various requests could be considered beyond the scope of the present study, with incorporation of your responses and appropriate qualification of certain conclusions being warranted instead. Again, improving the clarity and descriptions of Figure 3 shall be very important. Regarding their point about the EMSA in present Suppl. Fig 3C, it would be nice if you could add some discussion to the pros/cons of using EMSA to evaluate effects on open complex formation.

Editorial points (please, also refer to the Guidelines at the end of this email, and our online Guide to Authors, as sticking to them should greatly facilitate editorial processing at the time of resubmission):

- Please download (see link below) our author checklist, and upload it in completed form with the final manuscript.
- Please upload the manuscript text (including figure legends) as an editable text file, and all figures without legends as individual image files with sufficient resolution/quality for production.
- Please adjust the order of the manuscript sections: Title page with complete author information, Abstract, Keywords, Introduction, Results, Discussion, Methods, Data Availability, Acknowledgements, Disclosure and Competing Interests Statement, References, Main Figure Legends, Tables, Expanded Figure Legends.
- Please adjust the format of the reference list and of the in-text citations according to EMBO Journal format (alphabetical order, author name et al + year...)
- Please note that Materials and Methods need to be described in the main text using our 'Structured Methods' format (for detail,

see <https://www.embopress.org/page/journal/14693178/authorguide#structuredmethods>). The in-text "Methods" section should contain method and protocol descriptions (ideally using a step-by-step protocol format to facilitate adoption of the methodologies across labs), while all key reagents, experimental models, software and relevant equipment - including their sources and relevant identifiers - should be listed in a separately uploaded Reagents and Tools Table, a template for which can be downloaded from the above-linked section of our Author Guidelines.

- Please rename the "supplementary/extended figures" into Expanded View Figures (call-out: "Figure EV1/2/..."), both in the legends and when referencing them in the text. See www.embopress.org/page/journal/14602075/authorguide#expandedview for further information.

- Please rename the "supplementary tables" as Expanded View Table (reference: "Table EV1/2"), and upload them as a separate Word or Excel file, with their legend/title included.

- Please upload the "Extended Data Tables" mentioned in the text as "Expanded View Datasets" (callout "Dataset EV1/2") in Excel format, with their respective legends included in a separate "legends" tab of the spreadsheet. In accordance with our author guidelines (<https://www.embopress.org/page/journal/14602075/authorguide#datadeposition>), I would also encourage you to deposit the raw sequencing data in a relevant public database, such as GEO.

- Please rename the supplementary videos as Expanded View movies (in-text callouts again "Movie EV1/2/..."). Their legends should be moved out of the text into individual text files, each of which should be combined with the respective movie file into a separate ZIP file and uploaded as such.

- Finally, you shall also receive a separate message from our Source Data curation team, with instructions on how to prepare and upload relevant image and numerical raw data.

In this light, I am returning the manuscript to you now, in order to allow you to make these scientific and presentational revisions, and to upload all final files via the hyperlink below. Please, do not hesitate to get back to me with any remaining questions that you may have. Thank you again for the opportunity to consider this work for The EMBO Journal, and I look forward to receiving your revision!

With kind regards,

Hartmut

9) To facilitate reproducibility and cross-laboratory adoption of methodologies, please structure the Materials & Methods section as outlined in our guide to authors, including a completed Reagents and Tools Table that can be downloaded from our author guidelines as well (<https://www.embopress.org/page/journal/14602075/authorguide#structuredmethods>).

10) Digital image enhancement is acceptable practice, as long as it accurately represents the original data and conforms to community standards. If a figure has been subjected to significant electronic manipulation, this must be clearly noted in the figure legend and/or the 'Materials and Methods' section. The editors reserve the right to request original versions of figures and the original images that were used to assemble the figure. Finally, we generally encourage uploading of numerical as well as gel/blot image source data; for details see: embopress.org/page/journal/14602075/authorguide#sourcedata

At EMBO Press, we ask authors to provide source data for the main manuscript figures. Our source data coordinator will contact you to discuss which figure panels we would need source data for and will also provide you with helpful tips on how to upload and organize the files.

In the interest of ensuring the conceptual advance provided by the work, we recommend submitting a revision within 3 months (22nd May 2025). Please discuss the revision progress ahead of this time with the editor if you require more time to complete the revisions. Use the link below to submit your revision:

Link Not Available

25/03/2025

Dr Hartmut Vodermaier
Editor
The EMBO Journal

Graham R Stewart
Professor of Molecular Bacteriology
Bacteriology Section Leader
Department of Microbial Sciences
School of Biosciences
Faculty of Health and Medical
Sciences
University of Surrey
Guildford, Surrey GU2 7XH

T +44 (0)1483 686423
g.stewart@surrey.ac.uk

Article Submission/Revision Proposal:

Original Title: Epigenetic control of replication, gene expression and growth phase by ADP-ribosylation of DNA in *Mycobacterium tuberculosis*.

New Title: Control of replication and gene expression by ADP-ribosylation of DNA in *Mycobacterium tuberculosis*

Rachel E. Butler, Marion Schuller, Ritu Jaiswal, Jayanta Mukhopadhyay, Jim Barber, Suzie Hingley-Wilson, Alex Couto Alves, Emily Wasson, Ivan Ahel and Graham R. Stewart

Dear Dr Vodermaier,

Thank you for inviting a revised version of our manuscript. We have addressed all the points made by the reviewers following our proposed revision plan and your subsequent advice/direction. Thank you for your help, I think the paper is much improved. I've updated our point by point response to the reviewers and also your revision points (below). We have uploaded all the Figures, Expanded View Data/Tables/Movies/Figures/Synopsis and completed the Source Data. We look forward to hearing your thoughts on the revised manuscript.

Please don't hesitate to contact me if you have any questions.

Yours sincerely

Professor Graham R Stewart (and Ivan Ahel Co-corresponding author)

Scientific points from Editor:

- For referee 1, please incorporate the clarifications and presentational amendments, including the additional data/images, as outlined in your revision plan. I agree that Figure 1E may best be removed at this point.

We have added Figure EV1 (an additional quantitation of DNA and cfu in DarG knockdown BCG). We have added some Movie stills (Figure EV2) as suggested to go alongside Movies EV1/EV2. We have augmented Figure 1D to show the bacterial cell necrosis. We have removed Figure 1E. All other suggested changes from reviewer 1 have been made (see below).

- For Figure 3, I think that the labeling has to be improved (e.g., the legend says "lane 5, ... rrnA1-mut2 ... on the non-template strand" but "lane 6, rrnA1-mut2 ... of the template strand"?), although regarding the annotation of the three separate panels (and cf. ref. 3's related concern). This all makes it currently somewhat difficult to appreciate your response to major point 2 of referee 2, but do I understand correctly that you would propose to only retain lanes 1-4 & conclusions regarding the non-template strand in the final version? Please note that the additional replicates you mention for the template strand pausing/stalling, and/or quantifications, could easily be included as Expanded View or Appendix figures, with tempered conclusions if necessary.

We have changed the labelling for Figure 3 and updated explanations in the legend and the main text. We have also removed the Mut1 IVT template reactions in Figure 3B because they didn't add any additional understanding to the effect of ADP-ribosylation on transcription. Also for clarity, Figure 3C now shows only the EMSA for RNAP with the rrnAP3/ADPr DNA substrate (ADP-ribose on the non-template strand). We have toned down the conclusion that ADP-ribosylation of the template strand causes stalling of the RNAP although we still draw attention to this possibility and have added a repeat of the IVT in Figure EV6. We have also removed the stalling effect from the schematic on figure 5.

- Regarding the use of the contentious term "epigenetic", I am afraid I have to concur with referee 2 (and also referee 1) that it should only be used in the true sense of the meaning - to indicate traits being transmitted to daughter cells by means beside ("epi") the sequence-hardwired genetic inheritance (e.g., through passing on of a chromatin modification, of an RNA species, a prion protein...); but not as a collective term for every post-translational modification that reversibly affects gene expression. I would therefore appreciate removing the term from title, abstract, keywords and throughout the text, and instead switching to a more concrete terminology. I like the alternative title you suggest in response to referee 3, "Control of replication and gene expression by ADP-ribosylation of DNA in Mycobacterium tuberculosis."

Epigenetic is removed throughout the manuscript and the title updated.

- For referee 3, I appreciate that various requests could be considered beyond the scope of the present study, with incorporation of your responses and appropriate qualification of certain conclusions being warranted instead. Again, improving the clarity and descriptions of Figure 3 shall be very important. Regarding their point about the EMSA in present Suppl. Fig 3C, it would be nice if you could add some discussion to the pros/cons of using EMSA to evaluate effects on open complex formation.

Figure 3 has been updated as detailed above and below. For Fig3C we have added some extra discussion to read :

"We further investigated the effect of ADP-ribosylation of the template strand of rrnAP3 on the interaction with RNA polymerase using electromobility shift assay (EMSA). The result demonstrated that ADP-ribose modification did not inhibit the binding of the RNA polymerase holoenzyme to the promoter DNA (Figure 3C). Whether the observed inhibition of RNA transcript production from templates with ADP-ribosylation of the non-template strand adjacent to the -10 and -35 is due to an inability to form the open complex or an inability of

the RNA polymerase to escape from the ADP-ribose modified promoter needs further investigation because a heparin resistant shift is indicative but not proof of open complex formation."

Response to reviewers' comments.

Our response in blue and the yellow highlights changes made.

Referee #1: M. tuberculosis, physiology, DNA repair

Referee #2: ADP-ribosylation

Referee #3: M. tuberculosis biology

Reviewers Comments:

Reviewer #1 (Remarks to the Author):

The mechanisms available to genetically monomorphic pathogens such as *Mycobacterium tuberculosis* to maximize phenotypic diversity and phenotypic adaptation (despite possessing stable genomes without access to horizontal gene acquisition) are of considerable interest given the implications for pathogenicity and, potentially, chemotherapy. This latest study by Ahel, Stewart and colleagues builds on previous work by the same team and others characterizing the mycobacterial DarT/DarG toxin-antitoxin module which functions via reversible ADP-ribosylation of single-stranded genomic DNA. In the current submission, the authors extend the model by presenting evidence suggesting the activity of this module as an epigenetic regulator coordinating stationary phase growth with transcriptional adaptation – specifically demonstrating the capacity for ADP ribosylation-mediated control of the Zur regulon. This work provides some useful new insights into the role of DarT/G-dependent ADP-ribosylation in the physiology of mycobacteria, with potentially broader implications for other bacterial pathogens possessing homologous systems; however, the following should be considered:

1. Figure 1A repeats previous observations of the growth inhibitory effect of darG knockdown, but associates this with an impact on DNA synthesis (which is consistent with the inhibition of replication at OriC). Assuming the normalised DNA per well (right panel) is determined for the CFU/ml depicted in the left panel: why is there a difference between "Vector Control + ATC" (higher value) and "Vector Control", and why are the values for "darG sgRNA" and "darG sgRNA + ATC" not higher than those for the Vector controls given the ~1 log greater starting CFU values?

The fluorescence-based DNA assay used in this experiment is not sensitive enough to measure significant differences in the DNA content between strains at time-point 0 and 1. However, once the bacteria have undergone replication at time points 2 and 3 there is a significant difference in DNA concentration between the DarG-knockdown and the control

strains. Please also note that the apparent partial-discordance between cfu and DNA concentration profiles represents the fact that bacteria which undergo DNA replication do not necessarily undergo cell division. Indeed, this is the case for the DarG-knockdown bacteria which are only inhibited, but not completely blocked for DNA replication, but display a massive loss in the ability to form colonies on an agar plate.

2. Continuing from the above: Is it my understanding that these values were determined using the CyQuant Cell Proliferation Assay? If so, is there precedent for the use of this assay in mycobacteria (esp. *M. bovis* BCG) and a citation confirming its validation for this purpose; or can the authors provide additional supplementary information demonstrating the validation of the assay for this purpose?

Cyquant Direct is a well established dye-based assay for relative-quantification of the DNA content of cells. To back up use of this method we have included an additional figure (**Figure EV1**) showing OD600, cfu/ml and quantification of extracted DNA content by fluorometry in DarG-knockdown BCG and uninduced control BCG bacteria. The data corroborates the relationship between cfu/ml and extracted DNA observed in Figure 1A.

3. Additional evidence for the inhibition of DNA replication and cell division is apparently provided by the Supplementary Videos. However, I found these difficult to interpret, and of limited added value as presented. If they are to be included, my preference would be to see the raw videos augmented with a set of “still” images captured from the videos in a single additional (supplementary) figure, showing clearly the distinction between ATc-treated and ATc-unexposed cell extension and division. Incidentally, in this context, it’s unfortunate that the authors did not include a chromosomal marker (such as ParB [Santi & McKinney, 2015; doi:10.1128/mBio.01999-14] or equivalent) in the knockdown background to show clearly the impact of DarT/G dysregulation on chromosomal replication dynamics.

This is an excellent suggestion. We have produced a series of stills from the videos to highlight the differences in cell division/extension and include as Figure EV2 with Movies EV1 and 2.

4. In Figure 1B: How many single-cell measurements were captured for the data presented in the righthand panel? The legend indicates “three independent experiments” but it and the Methods section are silent (or opaque) on the total number of cells analysed.

We already stated in the methods that “A minimum of 39-60 bacilli were counted for each condition in n=2-3 replicate cultures. The average (Mean) of 3 independent experiments is presented.” However, we have now added to the figure 1B legend that each strain/treatment combination represents ~350 cell measurements, and we have updated the Methods to read “A minimum of 39-60 bacilli were counted for each condition in n=2-3 replicate cultures. The average (Mean) of 3 independent experiments is presented,

representing 358, 352, 362, and 348 cell measurements respectively for the vector control, vector control +ATC, darGsgRNA and darGsgRNA +ATC strains/treatments."

5. Necrosis is not evident in Fig. 1D as presented. As per my comment above, I would prefer to see a series of still images presented to accompany the videos; this seems critical given the importance of this observation for the subsequent experimental work and conclusions.

We have now Updated Figure 1D to show the necrosis and added an extended series of stills in Figure EV3 to go alongside Movie EV3.

6. The authors state "[w]e observed previously that the most obvious physiological phenotype of DarT-DarG epigenetic regulation is a slower growth rate...". This appears to be exaggerated: the cited study (Schuller; doi:10.1038/s41586-021-03825-4) demonstrated increased growth of the Mtb darT-darG knockout relative to its wild type parental strain, and also showed that transposon insertions in darT are abundant; there were no data presented, to my knowledge, in support of the claim that this effect resulted from "epigenetic regulation".

We have corrected the statement to read: "This corroborates observations that the presence of DarT-DarG confers a slower growth rate on wild-type *M. tuberculosis* compared to an isogenic DarT-DarG-knockout mutant (Schuller et al., 2021)."

7. The presentation of data in fig 1E is confusing: (i) why were the the specific timepoints (4d, 11d, 25-27d) selected? (ii) If DarT functions in "tolerance", as the Introduction and Discussion sections claim, then (a) why is the phenotype "lost" during extended exposure to RIF; and (b) strictly, demonstration of tolerance requires determination of Minimum Duration for Kill (MDK) values, as per Balaban (2019; doi: 10.1038/s41579-019-0196-3). Although onerous, there is precedent for the use of this assay in mycobacteria; e.g., Hicks (2018; doi:/10.1038/s41564-018-0218-3)

The altered rifampicin-sensitivity of the DarTG-knockout is indeed modest and does not represent one of the major findings of the paper. On reflection of the Balaban guidelines on mathematical definitions of antimicrobial tolerance as pointed out by the referee, we have removed figure 1E and reference to it in the text. Further experiments to more accurately define the nature of the altered sensitivity would be lengthy with *M. tuberculosis* and would not add significantly to the paper.

8. For the results presented in Figure 4, and Extended Data Table 2: The fact that no transcriptional differences were observed in log-phase cells seems surprising; given the fact that differences are, however, observed in stationary phase cells, is the interpretation that the level of darG/DarG is modulated in stationary phase cells?

Yes. This stationary phase dependency of DarT/G-regulation of the Zur-regulon is one of the main discoveries of this paper. The molecular mechanism of how this occurs is beyond the scope of the current paper, but likely involves the transcriptional linkage of DarT/G to the replicative helicase DnaB.

Minor comments

1. Introduction: The TB statistics (Ref 1) should be updated with the latest (2024) WHO Global TB Report.

Updated to WHO 2024 Global TB Report

2. The term "turn-off-and-on-able" is descriptive but clunky; could perhaps be replaced by equivalent such as "switchable", or "toggleable", or perhaps "dynamically adjustable" though that is less precise.

We have changed this to "dynamically flexible switch"

Reviewer #2 (Remarks to the Author):

The manuscript by Rachel Butler et al., "Epigenetic control of replication, gene expression and growth phase by ADP-ribosylation of DNA in *Mycobacterium tuberculosis*", describes reversible DarT/DarG-dependent ADP-ribosylation of DNA as a regulatory mechanism for replication and transcription. The study highlights how this process may contribute significantly to the pathogen's growth, adaptation, and survival. The authors build upon their prior work, which demonstrated that ADP-ribosylation of thymidine residues in ssDNA at the origin of chromosomal replication by DarT causes growth arrest in *Mycobacterium bovis* BCG. In this manuscript, they delve deeper into the mechanistic details of this regulation, map ADP-ribosylation sites across the genome of *M. bovis* BCG, and demonstrate the link between ADP-ribosylation and the regulation of replication and transcription of specific genes involved in pathogen adaptation. Overall, the manuscript is original, intriguing, and of high quality. However, the following points need to be addressed:

Major comments:

1. Appropriateness of the Term "Epigenetic":

The use of the term "epigenetic" in the title, abstract, and throughout the manuscript appears to be inappropriate. Epigenetic mechanisms traditionally imply mitotically heritable changes in gene expression, which has not been demonstrated in this study. To justify the term, the authors should provide evidence showing that ADP-ribosylation-induced changes are heritable across generations of the pathogen. Alternatively, the authors should revise the manuscript to avoid the use of "epigenetic" and adopt more precise terminology.

We have removed all use of "epigenetic" in the manuscript.

2. Clarification of RNA Polymerase stalling results (Figure 3B):

The authors state, "IVT of the Mut2 DNA substrate following ADP-ribosylation of the template strand, generated truncated RNA products up to 18 nt in length, correlating with the position of the engineered DarT motif, and very little full-length RNA product. This indicated that ADP-ribose modification of thymines on the template strand stalls RNA polymerase synthesis of RNA". However, the RNA product patterns shown in Figure 3B for both Mut2 and ADP-ribosylated Mut2 are strikingly similar. This suggests that ADP-ribosylation of the template strand has little to no impact on RNA polymerase holoenzyme activity, contrary to the conclusions drawn in the text. The authors should clarify this discrepancy and revise their conclusions accordingly.

We have simplified the *in vitro* transcription offering (fig 3B) in the revised manuscript and made the legend and labelling much clearer. We have removed the Mut 1 IVT reactions and relabelled the mut2 template as *rrnAP3+*. Of greatest relevance to the above reviewer comment, we have added a repeat of the IVT assay (Figure EV6) which provides some extra confidence in the repeatability of observed transcription changes, and we have toned down the inference that there is a transcriptional stalling effect of ADP-ribosylation of the template strand in the text and also removed it from the schematic in Figure 5. We now write "*IVT of the *rrnAP3+* DNA carrying ADP-ribosylation of the template strand exhibited no inhibition of the synthesis of abortive transcripts, although there was some evidence for stalling of the elongation complex near the introduced ADP-ribosylation sites. The latter requires further investigation.*"

Minor comments:

1. Figure 1E: Replace "DdarTDdar G" with " Δ darTdarG" in the figure.

Figure 1E has been removed from the manuscript

2. Figure 3B: Replace "WT F/T ADPr" with "WT F/R ADPr."

The labelling for Fig3 has been completely reworked to use Template (T) and Non-template (NT)

3. The abbreviation "OD600nm" is inconsistently formatted in the manuscript (e.g., "OD600nm" and "OD600 nm").

Amended

Reviewer #3 (Remarks to the Author):

Butler et al further investigate the DarTG toxin-antitoxin system in *M. tuberculosis* complex bacteria (MTBC). They make the quite interesting claim that DarTG functions as an epigenetic mechanism to coordinate stationary phase growth with transcriptional adaptation. Should this hypothesis prove to be true, this would be very interesting. However, this claim is unsubstantiated based on the provided data.

Major comments:

1. The primary claim of the paper is that “*M. tuberculosis* complex bacteria utilise reversible ADP-ribosylation of single-stranded DNA as an epigenetic mechanism to coordinate stationary phase growth with transcriptional adaptation.” Some aspects of this claim (e.g. the fact that DarT ADP-ribosylates DNA, as previously shown) are supported by the evidence. However, from this reviewer’s perspective, the most critical unresolved assertion is that, unlike its role in phage defense in other bacteria, DarTG has been adapted in MTBC to regulate stationary phase physiology. The experiments shown in this manuscript do not sufficiently support this claim. E.g. what is the endogenous trigger that liberates DarT in stationary phase?

We show very clearly that DarT/G are required to optimally regulate the Zur regulon in stationary phase and this is not observed in log phase. We demonstrate the mechanism of DarT-regulation of Zur-controlled genes using gel shifts to show that ADP-ribosylation specifically at the Zur binding site inhibits binding of Zur to the promoter. It is beyond the scope of this paper to show the trigger mechanism that coordinates the DarT activity with growth phase but it is highly likely to be associated with the fact that DarT and DarG are transcriptionally linked to the replicative helicase DnaB.

How is this trigger reversed to remove this modification to enable reentry into the cell cycle?

This is an interesting question but beyond the scope of this paper. We have previously published demonstration that DarG is the enzyme that removes ADP-ribose from thymidine and there is some data from other RNASeq studies (Cortes et al Cell Rep. DOI 10.1016/j.celrep.2013.10.031) that DarG has an alternative transcription start site which would allow differential expression from DarT, but exactly how this is regulated is beyond the scope of this paper.

If DarTG were critical to coordinate DNA replication with transcriptional adaptation to stationary phase, one might expect that DarT would be essential for *Mtb* pathogenesis in animal infection models- is this true?

Yes, if the experiment was done with a clinical strain of *Mtb* in a relevant animal model.

The only study to date (Zaveri et al *Mol Micro* 2020 doi: 10.1111/mmi.14571.) showed no difference in growth/survival between a DarT-knockout of the laboratory strain of *M.tuberculosis* H37Rv and wild-type in a mouse model of tuberculosis. The findings of our submitted paper (ie that DarT/G regulate the Zur-regulon) actually help us understand this negative result. Zur-regulon induction occurs in the caseum of the human TB granuloma (and in sputum) and does not overtly occur in mouse infection because the pathology of murine TB is completely different (no necrotic caseum and no zinc starvation). The only way we could feasibly compare wild-type and DarT-knockout in a relevant model would be to use cynomolgus macaques which do produce caseous granulomas/sputum. This is beyond the scope of our present project and it might be difficult to justify a primate experiment simply for this aim.

Without an answer to these sorts of questions, the most important claim of this paper remains unsubstantiated.

We disagree. In our previous paper (Schuller et al *Nature* 2021 doi.org/10.1038/s41586-021-03825-4) we demonstrated that DarT-dependent ADP-ribosylation of DNA OriC is important for control of replication. Here, we provide extra characterisation of this mechanism, and for the first time in any organism, we characterise a genome-wide DNA-ADP-ribosylome, showing extensive ADP-ribosylation at and around OriC but also ADP-ribosylation across the genome. We further demonstrate that ADP-ribosylation of the Zur binding site dysregulates expression of the ZUR-regulon in stationary phase bacteria and that ADP-ribosylation of DNA can also inhibit the activity of RNA polymerase.

However, we acknowledge that we only show that darT/G controls one aspect of stationary phase adaptation and thus the paper title was perhaps too bold.

We have amended the title to reflect this: "**Control of replication and gene expression by ADP-ribosylation of DNA in *Mycobacterium tuberculosis*.**"

2. Related to point 1 above, the authors cite the differential expression of 33 genes in darTG-KO as compared to WT as key evidence that DarTG is regulating stationary phase biology in Mtb. Are these 33 genes shown to be ADP-ribosylated in their sequencing results in Figure 2 (and in the associated Supplemental Table)?

No, the 33 differentially expressed genes in stationary phase DarTG-knockout bacteria (Figure 2) were not ADP-ribosylated in the DarG-knockdown bacteria in log phase (Figure 3). We assume this is because the bacteria are in different growth phases and undergoing different genetic perturbations, and the ADP-ribosylome is dynamically changing dependent upon replication state and the differing availability of ssDNA DarT substrate at replication and transcriptional sites. We are presently working on the sensitivity of the ADPr-Seq method with a view to mapping the DNA-ADP-ribosylome in natural growth states; here we provide a first

demonstration (in any organism) of ADPr-Seq using DarG-knockdown bacteria which have unregulated DarT activity.

Conversely, how do the authors explain that the genes they do find to be ADP-ribosylated in their sequencing results are not differentially expressed in their RNAseq experiment?

For the same reasons as above.

3. Related to point 2, the RNAseq experiment is key to the authors argument. How many times was this experiment performed? Have the authors confirmed these expression differences by qPCR in an independent experiment?

The RNA-Seq experiment comparing stationary phase DarTG-knockout and wild-type *M.tuberculosis* (in Figure 4A) shows the mean values of 4 independent replicate cultures for each strain where each replicate culture has been grown independently for 20 days. We validated the specific involvement of DarTDarG in regulating the Zur-regulon with a second set of RNA-Seq comparisons using the zinc chelator TPEN at levels which limited zinc but did not inhibit growth of the mycobacteria. We observed that the *M. tuberculosis* Δ darTdarG was impaired in its ability to induce Zur-regulated genes although at levels significantly lower than observed in stationary phase (Figure 4D, Dataset EV2). In combination with the biochemical assays showing the effect of DarT-dependent ADP-ribosylation on Zur-binding sites, we thus provide several different types of experimental evidence, each with independent replicates, which all indicate that DarTDarG has a regulatory effect on the Zur-regulon by ADP-ribosylation of DNA at the Zur-binding site.

Have the authors validated by whole genome sequencing that their strains are isogenic except for the darTG-deletion?

Yes. We were exceptionally careful to provide isogenic mutant and wild-type strains for comparison. We derived the parent *M. tuberculosis* GC1237 from a single colony and passaged only twice before performing gene replacement to make the DarTDarG mutant. Both strains were sequenced to verify the gene replacement and the absence of secondary mutations. This is now more clearly stated in the revised manuscript.

4. Figure 1E is not convincing that darTG-KO die faster in rifampicin. The effect size is quite modest. How many times was this experiment performed? Is this phenotype observed for other antibiotics?

We agree that this is a modest effect although it is highly repeatable. However, given that it doesn't form the central focus of the study and in response to reviewer #1, we have removed Figure 1E from the revised manuscript.

5. Fig. 3A: Please use consistent terminology throughout the figure, referring to "Template" (T) and "Non-Template" (NT) strands instead of "F" and "R." Additionally, clearly label which strand

corresponds to T and NT for the readers benefit. Highlight the bases that are putatively ADP-ribosylated in this oligonucleotide mapping experiment.

We have made these changes to Figure 3.

Furthermore, are these experiments examining ribosylation of single-stranded DNA? If so, is this representative of how DarT operates in vivo? It seems likely that DarT should interact primarily with double-stranded DNA.

We have previously demonstrated that the ADP-ribosylation activity of DarT is specific for ssDNA (Jankevicius et al *Mol Cell* [10.1016/j.molcel.2016.11.014](https://doi.org/10.1016/j.molcel.2016.11.014)) and we state this clearly in the second paragraph of the Introduction in the submitted manuscript. The ssDNA-dependence of DarT activity is probably an important feature of its specificity for the chromosome origin of replication and also transcriptionally active genes (like Zur in stationary phase or under zinc depletion) where ssDNA is exposed.

6. Fig. 3 B (first two panels): The first two panels are unclear. Please provide a more detailed description in the figure legend.

We have improved the labelling in a revised Figure 3 and added to the legend.

If I understand correctly, the panels depict strands believed to be modified (+/- DarT). However, to validate these claims, both strands (+/- DarT) should be shown to confirm the absence of modifications on the complementary strand.

This is not necessary. The reviewer has misunderstood these experiments. The individual strands have been ADP-ribosylated by reaction of the ssDNA with DarT in the presence of B-NAD⁺. Figure 3B left panel demonstrates the successful modification of the ssDNA. These ssDNA are then annealed to their unmodified complementary strands (these are shown in Figure 3 B middle panel). We have improved the legend text and the main text to make sure that this is clear.

Moreover, the lack of a molecular weight ladder makes it impossible to determine the lengths of the DNA fragments. Including a schematic diagram illustrating the experimental setup and indicating the positions of modifications in Mut1 and Mut2 would improve the clarity of this figure.

The sequence (and with that the length) of the DNA promoter fragments is provided in the schematics of Figure 3A but we have improve the labelling and the legend. Due to the small size of ADP-ribose (MW: 559 g/mol), the size difference between the unmodified and ADP-ribosylated strand cannot be judged using a MW weight ladder for DNA. Instead, we therefore use internal referencing for judging successful ADP-ribosylation of the DNA

template: The negative (-) controls are samples which have been treated as all other samples but have buffer instead of DarT added to the reaction, thus showing the unmodified oligo for internal referencing to the ADPr-modified oligos. The positive (+) controls are samples which have also been treated as all other samples and have DarT added to the reaction, thus showing the ADPr-modified oligo for internal referencing to the unmodified oligos.

The sequence of IVT DNA templates is provided in Table EV2 and stated in the Methods. We have added nucleotide coordinates (-59 to +47) to the figure and legend and the position of the base change (+23 g to t) in *rrnAP3+* template (was Mut2) is stated.

7. Fig. 3B: The gel results are perplexing. Why does the double-stranded DNA (dsDNA) exhibit faster migration (indicating lower mobility) than the single-stranded DNA (ssDNA)? Again, the absence of molecular weight ladders hinders interpretation.

In general, double-stranded DNA (dsDNA) migrates faster than single-stranded DNA (ssDNA) in gel electrophoresis because the dsDNA structure is more rigid and compact due to hydrogen bonding between base pairs, allowing it to move through the gel matrix with less resistance compared to the flexible, more convoluted ssDNA which encounters more friction with the gel pores. Furthermore, dsDNA has a higher "effective charge density" leading to faster migration in gel electrophoresis. However, ADP-ribosylation of the strands alters the secondary structure of the dsDNA, including by interfering with base-pairing around the ADP-ribose attachment site. This makes the ADPr-dsDNA product less compact and rigid to run as fast as unmodified dsDNA through the gel matrix and the migration behaviour compares to the ssDNA as observed. Nevertheless, dsDNA and ssDNA products can be easily distinguished by the higher strength of ethidium bromide staining of dsDNA products due to stronger dye complex formation between complementary DNA strands, resulting in the noticeable broader staining pattern as compared to the corresponding ssDNA products.

<https://www.sciencedirect.com/science/article/pii/S0006349520303489>

Is the WT "F" strand already ADP-ribosylated prior to the experiment?

See above

Additionally, the figure itself lacks lane numbers, despite their reference in the description, which further complicates interpretation.

The labelling has been made much clearer.

- Fig. 3 (right panel): The absence of ladders here also makes the figure difficult to interpret. The transcription assays (IVT) show substantial issues:

We don't consider that a ladder would be necessary because one can count the nucleotides from nucleotide 1 at the bottom of the gel. Key sizes are indicated.

- There is an excessive amount of abortive products relative to run-off products-the assays need optimization.

The *rrnAP3* promoter is one of the most studied mycobacterial promoters (eg. <https://doi.org/10.1016/j.jbc.2023.104724> ; <https://doi.org/10.1093/nar/gks346>) and we chose it because it has already been characterised in the literature and because the sequence around the -10 matched the preferred target sequence for DarT-dependent ADP-ribose modification. This promoter inherently produces a high level of abortive transcripts but there is significant full-length transcript produced in our assays, and indeed we have a stronger "runoff" band than demonstrated in other studies (eg. <https://doi.org/10.1093/nar/gks346>). "Optimisation" will not change the characteristics of this template, which functions very well in the IVT assay to show the effect of ADP-ribose modification at the -10 site.

- If ADP-ribosylation at the NT -10 position inhibits initiation (as the data suggest), why is there transcription from Mut2, which presumably retains ADP-ribosylation at the -10 position?

No, the MUT2 template only has ADP-ribose modification on the template strand (generated by exposure to DarT/B-NAD⁺ as explained above). The non-template strand containing the ADP-ribosylation sites at the -10 has not been modified.

- Instead of using EMSA in Supplementary Fig. 3C, perform a potassium permanganate assay to directly evaluate whether ADP-ribosylation affects open complex formation.

This is a good suggestion and we will consider this in future characterisation experiments but are unable to perform this presently. We have now clarified our conclusion from the EMSA: "We further investigated the effect of ADP-ribosylation of the template strand of *rrnAP3* on the interaction with RNA polymerase using electromobility shift assay (EMSA). The result demonstrated that ADP-ribose modification did not inhibit the binding of the RNA polymerase holoenzyme to the promoter DNA (Figure 3C). Whether the observed inhibition of RNA transcript production from templates with ADP-ribosylation of the non-template strand adjacent to the -10 and -35 is due to an inability to form the open complex or an inability of the RNA polymerase to escape from the ADP-ribose modified promoter needs further investigation because a heparin resistant shift is indicative but not proof of open complex formation."

In conclusion, regarding the transcription assays: As currently presented, the transcription assays are confusing and do not convincingly substantiate the authors' claims. While I agree that ADP-ribosylation likely inhibits initiation (rather than promoter escape as suggested by the authors), the mechanism remains unclear. Specifically, it is confusing how ribosylated Mut2 can serve as a template for abortive transcription given that it carries the same promoter modifications as WT and Mut1. These discrepancies should be addressed, and the assays require optimization to provide more definitive conclusions.

We have revised the figure to make it clearer.

Minor comments:

1. The first time this reviewer read about the function of DarTG in Mtb was in a 2020 paper from the Schnappinger lab (PMID: 32634279). It is not clear why the Schnappinger paper is so infrequently cited in the current manuscript. There are many instances (e.g. paragraph starting on line 51, line 71, etc) in which PMID: 32634279 should be cited, in addition to the current authors own work (PMID: 34408320).

We think we already cite the Zaveri et al (Schnappinger lab) paper appropriately.

2. Line 68: do MTBC have defence islands?

They are not characterised as such, nor studied, but the genomic organisation is suggestive. A future project perhaps.

3. Line 106: it is not clear to this reviewer that necrosis has been described in Mtb.

We use this term to describe a lytic form of cell death.

4. The Discussion would benefit from an expanded discussion of what determines DarT specificity in the genome? Why are only some sites with the necessary recognition site targeted?

We already have the following paragraph on this issue: "ADP-ribosylation of thymidine nucleosides by DarT is restricted by DNA target sequence and also the occurrence and accessibility of ssDNA. The OriC itself has multiple high-efficiency DarT target sequences in the AT-rich region which functions primarily as the initial site for ssDNA generation allowing loading of the DnaB helicase (co-transcribed with darT-darG) and the DNA polymerase which further unwind and replicate DNA. It is thus clear that this region has the biochemical and physiological characteristics that explain the high-level of ADP-ribosyl modification observed. We postulate that ADP-ribosylation of thymines outside of the OriC region may be partly-dependent on transcriptional activity to provide the ssDNA template for DarT."

Prof. Graham Roger Stewart
University of Surrey
School of Biosciences
Guildford, SURREY GU27XH
United Kingdom

2nd Apr 2025

Re: EMBOJ-2025-120334R
Control of replication and gene expression by ADP-ribosylation of DNA in *Mycobacterium tuberculosis*

Dear Dr. Stewart,

Thank you for submitting your revised manuscript to The EMBO Journal. I have now carefully checked your responses to my previous decision letter, as well as your updated responses to the transferred original comments and their incorporation into the final version. I am happy with all scientific revisions, but there remain still a few editorial issues to be addressed before we can proceed with formal acceptance and production of the article:

- The order of the manuscript sections still needs to be fully adjusted as follows: Title page with complete author information, Abstract, Keywords, Introduction, Results, Discussion, Methods, Data Availability, Acknowledgements, Disclosure and Competing Interests Statement, References, Main Figure Legends, Tables, Expanded Figure Legends.
- Please reduce the number of keywords on the abstract page to five, ideally preferring general conceptual terms over e.g. gene names or non-unique abbreviations like TB
- Please double-check to make sure to all relevant funding information in the manuscript is congruent with the info entered into our submission system (the Guy Newton Translation Fund (grant GN05 18); the Goodger and Schorstein Scholarships Trust Fund (Source of funds BC076) appear to be missing in the submission system).
- Please carefully check the bibliography for completeness (journal, year, volume, pagination...) of all references; especially page/locator numbers seem to be missing in several instances.
- For the EV figures, please make sure to not only reference them by figure number, but to also call out their individual panels at least once (e.g., by referring to Figure EV1A-C instead of Figure EV1). Also, for Fig EV2, individual panels A-F are mentioned in the figure legend, but are not indicated in the figure - please adjust either in the image or in the legend. Finally, the legend for Fig EV3 mentions all panels except panel F - please correct.
- For Datasets EV1/2 and Table EV2, please add in each case a separate "legends" tab in the spreadsheet, which should contain the number/header/title of the respective item, as well as any legend information required.
- Finally, during routine pre-acceptance checks, our data editors raised that the exact P values are not provided in the legends of figures 1B, EV1 B, C; EV5 - please correct.

I am therefore returning the manuscript to you for a final round of minor revision, to allow you to make these modifications and upload the revised files. Once we will have received them, we should be ready to swiftly proceed with formal acceptance and production of the manuscript.

Yours sincerely,

Hartmut Vodermaier

- 1) Every manuscript requires a Data Availability section (even if only stating that no deposited datasets are included). Primary datasets or computer code produced in the current study have to be deposited in appropriate public repositories prior to resubmission, and reviewer access details provided in case that public access is not yet allowed. Further information: embopress.org/page/journal/14602075/authorguide#dataavailability
- 2) Each figure legend must specify
 - size of the scale bars that are mandatory for all micrograph panels
 - the statistical test used to generate error bars and P-values
 - the type error bars (e.g., S.E.M., S.D.)
 - the number (n) and nature (biological or technical replicate) of independent experiments underlying each data point
 - Figures may not include error bars for experiments with $n < 3$; scatter plots showing individual data points should be used instead.
- 3) Revised manuscript text (including main tables, and figure legends for main and EV figures) has to be submitted as editable text file (e.g., .docx format). We encourage highlighting of changes (e.g., via text color) for the referees' reference.
- 4) Each main and each Expanded View (EV) figure should be uploaded as individual production-quality files (preferably in .eps, .tif, .jpg formats). For suggestions on figure preparation/layout, please refer to our Figure Preparation Guidelines: <http://bit.ly/EMBOPressFigurePreparationGuideline>
- 5) Point-by-point response letters should include the original referee comments in full together with your detailed responses to them (and to specific editor requests if applicable), and also be uploaded as editable (e.g., .docx) text files.
- 6) Please complete our Author Checklist, and make sure that information entered into the checklist is also reflected in the manuscript; the checklist will be available to readers as part of the Review Process File. A download link is found at the top of our Guide to Authors: embopress.org/page/journal/14602075/authorguide
- 7) All authors listed as (co-)corresponding need to deposit, in their respective author profiles in our submission system, a unique ORCID identifier linked to their name. Please see our Guide to Authors for detailed instructions.
- 8) Please note that supplementary information at EMBO Press has been superseded by the 'Expanded View' for inclusion of additional figures, tables, movies or datasets; with up to five EV Figures being typeset and directly accessible in the HTML version of the article. For details and guidance, please refer to: embopress.org/page/journal/14602075/authorguide#expandedview
- 9) To facilitate reproducibility and cross-laboratory adoption of methodologies, please structure the Materials & Methods section as outlined in our guide to authors, including a completed Reagents and Tools Table that can be downloaded from our author guidelines as well (<https://www.embopress.org/page/journal/14602075/authorguide#structuredmethods>).
- 10) Digital image enhancement is acceptable practice, as long as it accurately represents the original data and conforms to community standards. If a figure has been subjected to significant electronic manipulation, this must be clearly noted in the figure legend and/or the 'Materials and Methods' section. The editors reserve the right to request original versions of figures and the original images that were used to assemble the figure. Finally, we generally encourage uploading of numerical as well as gel/blot image source data; for details see: embopress.org/page/journal/14602075/authorguide#sourcedata

At EMBO Press, we ask authors to provide source data for the main manuscript figures. Our source data coordinator will contact you to discuss which figure panels we would need source data for and will also provide you with helpful tips on how to upload and organize the files.

In the interest of ensuring the conceptual advance provided by the work, we recommend submitting a revision within 3 months (1st Jul 2025). Please discuss the revision progress ahead of this time with the editor if you require more time to complete the revisions. Use the link below to submit your revision:

Link Not Available